# Mechanical design principles of a mitotic spindle

Jonathan J Ward[1][†], Hélio Roque[1,2][†], Claude Antony[1,3], François Nédélec[1]*

[1]Cell Biology and Biophysics Unit, European Molecular Biology Laboratory, Heidelberg, Germany; [2]Sir William Dunn School of Pathology, University of Oxford, Oxford, United Kingdom; [3]Department of Integrated Structural Biology, UMR7104, Institut Génétique Biologie Moléculaire Cellulaire, Illkirch, France

**Abstract** An organised spindle is crucial to the fidelity of chromosome segregation, but the relationship between spindle structure and function is not well understood in any cell type. The anaphase B spindle in fission yeast has a slender morphology and must elongate against compressive forces. This 'pushing' mode of chromosome transport renders the spindle susceptible to breakage, as observed in cells with a variety of defects. Here we perform electron tomographic analyses of the spindle, which suggest that it organises a limited supply of structural components to increase its compressive strength. Structural integrity is maintained throughout the spindle's fourfold elongation by organising microtubules into a rigid transverse array, preserving correct microtubule number and dynamically rescaling microtubule length.

*For correspondence: nedelec@embl.de

[†]These authors contributed equally to this work

Competing interests: The authors declare that no competing interests exist.

## Introduction

It has long been proposed that the morphology of organisms could reflect both evolutionary design principles and underlying physical laws (*Thompson, 1942*). The organelle responsible for faithful segregation of the genome in eukaryotic cells, known as the mitotic spindle, exhibits tremendous morphological diversity between different cell types (*Walczak and Heald, 2008*; *Glotzer, 2009*; *Wühr et al., 2009*; *Goshima and Scholey, 2010*). Although the mitotic spindle has been studied intensively for more than a century (*Mitchison and Salmon, 2001*), our understanding of the mechanisms that give rise to the remarkable precision of eukaryotic chromosome segregation remains incomplete.

A challenge facing mechanistic studies of the mitotic spindle is its inherent complexity, with a large number of essential protein components (*Walczak and Heald, 2008*; *Glotzer, 2009*) and an intricate, extended structure with many details that cannot be visualised by light microscopy (*McDonald et al., 1992*; *Mastronarde et al., 1993*). This complexity is compounded by the substantial morphological and temporal variability that often exists between different cells of the same type. Another complication is that the spindle is usually involved in performing multiple interrelated functions in the cell at any one time. During prometaphase, for example, the spindle must simultaneously capture mitotic chromosomes and form a bipolar structure (*Walczak and Heald, 2008*; *Glotzer, 2009*). We chose the anaphase B spindle in fission yeast as a model system to address the relationship between spindle form and function as it circumvents many of these conceptual difficulties.

The first advantage of studying mitosis in *Schizosaccharomyces pombe* is that the rate of elongation and the timing of each mitotic stage are highly consistent in different cells (*Mallavarapu et al., 1999*; *Fu et al., 2009*). This enables the dynamics of mitosis to be related directly to the comprehensive, nano-scale reconstructions of the spindle architecture that can be acquired using electron tomography. A second advantage is that anaphase B spindle elongation represents a clearly defined morphogenetic transition that is driven by outward sliding of overlapping microtubules by

**eLife digest** Before a cell divides to form two new cells, it duplicates its entire set of chromosomes. These chromosomes need to be equally distributed between the new cells—if cells receive too many or too few chromosomes, it can cause developmental defects or cancer.

In cells that have a nucleus, a structure called the mitotic spindle ensures that chromosomes are partitioned equally between the dividing cells. The spindle consists of long, thin protein fibers called microtubules, which grow from small structures known as centrosomes that are present on either side of a cell. While some of the microtubules from each centrosome overlap in the middle of the spindle in a region called the spindle midzone, another set of microtubules attaches to the chromosomes, allowing the spindle to pull each of the chromosomes in a pair in opposite directions.

The size and shape of the mitotic spindle varies widely between different species, and how the structure of the spindle helps it to do its job was unclear. However, it is known that the spindle has to be strong and fairly rigid in order to separate the chromosomes.

Ward, Roque et al. studied the chromosome separation process in a species of yeast that has unusually consistent growth and cell division rates in different cells. In a technique called electron tomography, an electron microscope took images of the spindle from many different angles, and these images were combined computationally to produce a three-dimensional structure of the entire spindle.

Ward, Roque et al. observe that the number and length of microtubule fibers in the spindle is the same in each yeast cell. The spindle also has a striking geometric pattern. In the spindle midzone, microtubules are ordered into a highly regular square-packed array, while the rest of the spindle contains microtubules arranged hexagonally. This hexagonal arrangement maximizes the interactions between a microtubule and its neighbors, which makes the spindle stronger and prevents it from buckling under the physical forces that act on it. Engineers have incorporated this type of design in man-made structures for decades. A future challenge is to explain how the properties of the spindle components have been tuned to be able to always assemble into a structure with such reliable properties.

plus-end directed molecular motors located at its centre (*Tolić-Nørrelykke, Sacconi, et al., 2004b*; *Khodjakov et al., 2004*; *Fu et al., 2009*; *Glotzer, 2009*). Microtubule minus-ends are static and remain anchored at the spindle pole body, whilst growth of inter-polar microtubules at their plus-ends is coordinated with outward sliding to maintain an overlap at the spindle centre (*Mallavarapu et al., 1999*).

A final key advantage of studying the anaphase B spindle is that the forces to which it is subjected have a well-defined directionality. Spindle severance experiments have revealed that spindle elongation is powered by internal forces, and resisted by external compressive loads (*Tolić-Nørrelykke, Sacconi, et al., 2004b*; *Khodjakov et al., 2004*). These external loads can compromise the fidelity of chromosome segregation, as it has been observed that buckling of the spindle followed by its breakage is a common failure mode for cells that have defects in chromosome condensation, the nuclear envelope or in the organisation of spindle microtubules (*Courtheoux et al., 2009*; *Khmelinskii et al., 2009*; *Yam et al., 2011*; *Petrova et al., 2013*).

In this study, we reconstruct the architecture of wild-type fission yeast spindles using electron tomography (ET) (*Höög et al., 2007*; *Roque et al., 2010*). This technique has several advantages (*Soto et al., 1994*) over the serial-section electron microscopy method that was used previously to determine the structures of spindles in cdc25.22 fission yeast and budding yeast cells (*Ding et al., 1993*; *Winey et al., 1995*). We use the analyses of the EM spindle reconstructions to build computational models of the spindle. These models imply that the fission yeast spindle architectures organise a limited supply of structural components to increase their resistance to compressive forces, thus demonstrating a direct link between the morphology of a mitotic spindle and its function. We also investigate the effects of external mechanical reinforcement (*Pickett-Heaps et al., 1997*; *Mitchison et al., 2005*; *Brangwynne et al., 2006*) on the forces that the spindle can bear during its elongation.

## Results

### Electron tomographic reconstructions of wild-type Fission Yeast Spindles

We began by using the very uniform mitotic progression of cells containing GFP-labelled tubulin (*Figure 1A*; *Video 1, 2*) to assign ET reconstructions to a specific time during mitosis. Our ET reconstructions confirmed that the spindle is composed of two opposing arrays of pole-nucleated microtubules that interdigitate at the spindle midzone (*Figure 1B*; *Videos 3–5*). A similar gross spindle organisation was observed in previous serial-section reconstructions of cdc25.22 fission yeast cells (*Ding et al., 1993*) and the related budding yeast spindle (*Winey et al., 1995*).

In order to obtain estimates of the spindle's cross-sectional (or transverse) organisation that are independent of any spindle distortions (*Figure 1B*; see subsequent discussion), we developed a new algorithm, termed Isotropic Fibre Tracking Analysis (IFTA). This procedure enabled us to determine the spindles' contour length and to investigate the transverse organisation of microtubules ('Materials and methods'). A visualisation of the IFTA results (*Figure 1C*) reveals that microtubules at the spindle midzone form arrays with a degree of square-packed order from late metaphase onwards. We frequently observe square arrays containing a 3 × 3 'checkerboard' arrangement of microtubules at the midzone of spindles from the end of metaphase and early in anaphase B, whilst all of the later spindles contain at least four microtubules arranged as the vertices of a square. The polar regions of the spindle are loosely packed in metaphase but become more tightly packed in anaphase B, where we observe three microtubules arranged in an equilateral triangle motif in all of the reconstructed spindles.

An inspection of the raw tomographic images (*Figure 1D*) reveals good agreement with the results of IFTA, and also shows that thin filaments of electron density that bridge the spaces between microtubules are present at both the midzone and the flanking regions. An analysis of the lengths of microtubules with respect to the spindle's contour length (*Figure 1C*) shows that the microtubule overlap spans across the spindle in metaphase. In anaphase B, the midzone occupies the central portion of the spindle with a symmetric width of around 2 µm.

Statistical analyses of IFTA results from anaphase B spindles (*Figure 1E*) confirm the square-packed character (θ ~ 90°) with a large spanning distance (d ~ 40 nm) at the spindle midzone. Conversely, denser packing (d ~ 30 nm) with hexagonal character (θ ~ 60°) is present in the regions that flank the midzone (*Figure 1F*). The square packing suggests that the microtubules are bundled in an anti-parallel orientation at the spindle midzone (*Ding et al., 1993*; *Janson et al., 2007*); whilst hexagonal packing in the regions closer to the spindle poles is indicative of microtubules being bundled in parallel (*McDonald et al., 1979*).

A prominent feature of the ET reconstructions is the relatively small number of constituent microtubules. The ET shows that the spindle contains around thirty microtubules during metaphase, and that this number is reduced dramatically by kinetochore-fibre depolymerisation in anaphase A (*Figure 1G*). Upon entry into anaphase B, the spindle is constructed from only ten microtubules, which are then lost progressively from the spindle until only six remain prior to its disassembly. Concomitant with the changes in microtubule number, the total microtubule length polymerised within the spindle peaks at around 30 µm at the end of metaphase, but then remains roughly constant in anaphase B as the spindle length increases by a factor of four (*Figure 1H*). This result was confirmed by live-cell imaging, which indicated that the intensity of the GFP-labelled tubulin incorporated into the spindle also plateaus shortly after the metaphase-to-anaphase transition (*Figure 1H*; *Figure 1—figure supplement 1*; *Videos 1 and 2*). This analysis also affirmed the highly stereotypic spindle elongation profile in fission yeast cells (*Figure 1I*).

The close agreement between the live-cell imaging and static ET enabled us to estimate the critical and total concentration of tubulin subunits in fission yeast cells by calibrating the fluorescent intensity measurements to the total tubulin polymerised in ET reconstructions of the spindle (*Table 1*). This technique yields an estimate for the tubulin concentration (4.3 ± 0.8 µM) that is a factor of five lower than in metazoan cells (*Gard and Kirschner, 1987*), and is in good agreement with mass spectrometry estimates of the abundance of tubulin isoforms (*Marguerat et al., 2012*). Since all microtubule dynamics occur at the plus-tips of spindle microtubules during anaphase B (*Mallavarapu et al., 1999*), the conservation of polymer mass suggests that microtubule depolymerisation and recycling of tubulin subunits into the free pool (*Walker et al., 1988*) may be required for growth of the surviving interpolar microtubules, as suggested by previous EM reconstructions of fission yeast cells containing the

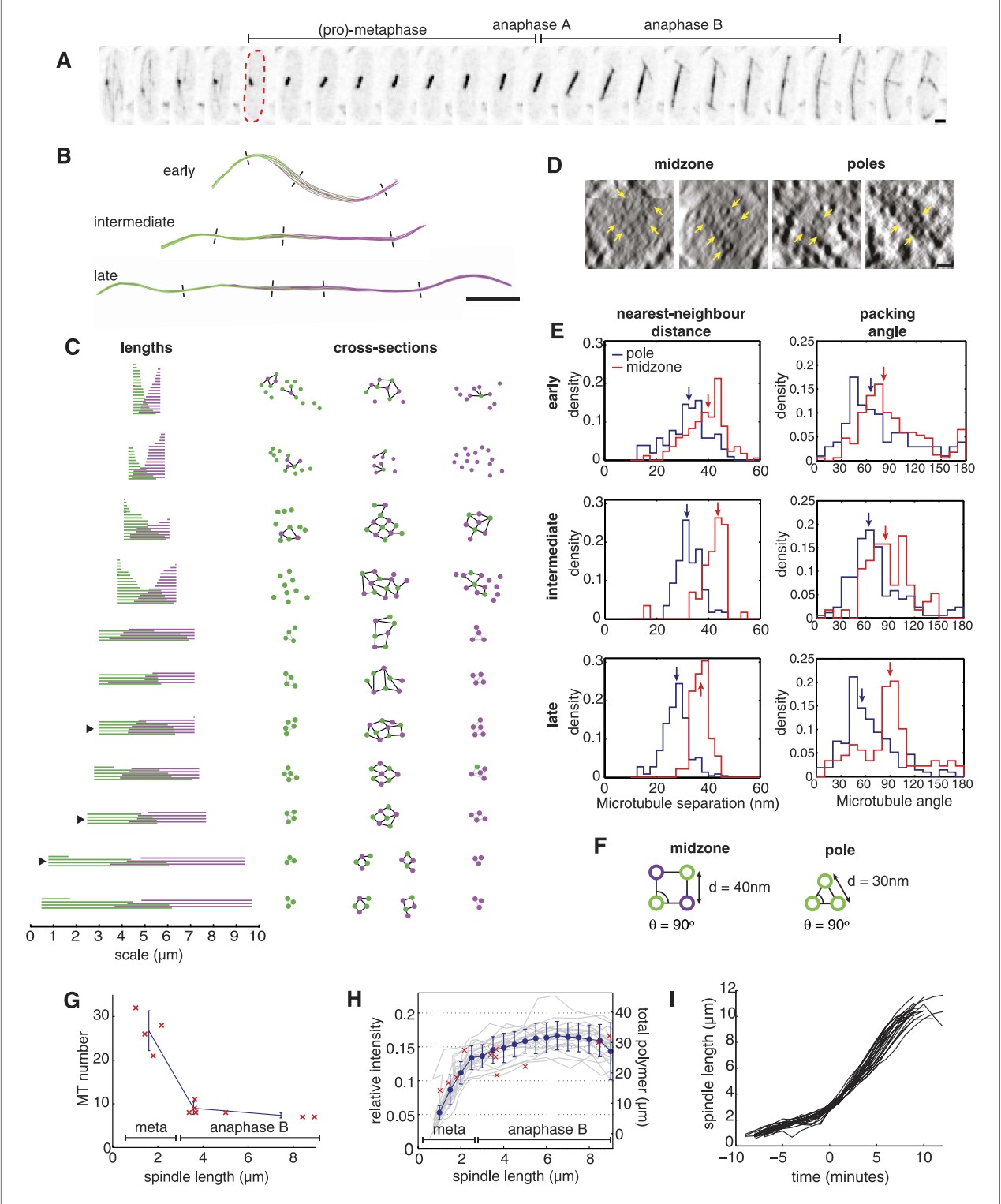

**Figure 1**. The Architecture and Dynamics of the Fission Yeast Spindle. (**A**) Fission yeast cell expressing GFP-labelled tubulin. The dashed red line shows cell outline at mitotic entry. Interval between frames = 1 min, scale bar = 0.5 μm. (**B**) ET reconstructions of three anaphase B spindles. Microtubules are coloured according to the pole from which they originate. Dashes represent the approximate locations of the cross-sections shown in **C**. (**C**) Longitudinal and transverse spindle architecture. Diagrams on the left show the contour length and number of microtubules within each spindle. Black arrowheads mark the three anaphase B spindles depicted in **B**. Circle-and-stick diagrams on the right represent cross-sections near to the poles and midzone of each
*Figure 1. Continued on next page*

*Figure 1. Continued*

spindle. Microtubule radii are drawn to scale. Black lines are drawn between neighbouring anti-parallel microtubules. (**D**) Raw electron tomographic images of second-longest anaphase B spindle shown in **C**. Microtubules are marked by yellow arrows. Scale bar = 25 nm. (**E**) Histograms of nearest-neighbour microtubule distances and packing angles for the three spindles shown in panel **B**. Blue lines represent histograms for the polar regions of the spindle and red lines the midzone. Coloured arrows mark the median of each distribution. The midzone is defined as the region of microtubule overlap that contains at least two microtubules from each pole. Kolmogorov–Smirnov tests were used to determine the probability that distributions for the polar and midzone regions were drawn from the same parent. All of the comparisons, shown here, have a p-value <10⁻⁴. (**F**) Schematic representation of microtubule angle and distance distributions at the poles and midzone of the spindle. (**G**) Number of spindle microtubules with respect to pole-to-pole spindle length $L_s$. The blue error bars summarise results for spindles divided into metaphase ($L_s < 2.5$ μm), early anaphase B ($2.5 < L_s < 4$ μm), and late anaphase B ($L_s > 4$ μm) stages. (**H**) Estimates of the ratio of fluorescent tubulin incorporated into the spindle for twenty mitotic cells (grey curves) with mean and standard deviations shown in blue. The red crosses indicate the total length of microtubules polymerised within each ET spindle. (**I**) Elongation profile for twenty wild-type spindles obtained from automatic tracking of SPBs. The traces are aligned temporally by defining time-zero as the frame at which spindles first reach a length of 3 μm.

The following figure supplement is available for figure 1:

**Figure supplement 1**. Live-cell imaging confirms that the mass of tubulin polymerised in the spindle is conserved throughout anaphase B.

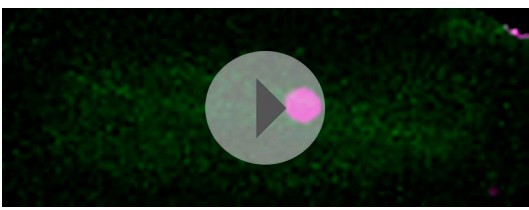

**Video 1**. Spindle formation and elongation in fission yeast. Frames are shown at intervals of 1 min. The green channel shows maximum intensity projections of cells expressing GFP-tubulin (SV40:GFP-Atb2), with the magenta channel showing SPBs labelled with Cut12-tdTomato. The cut12-tdTomato images were processed using a deconvolution algorithm.

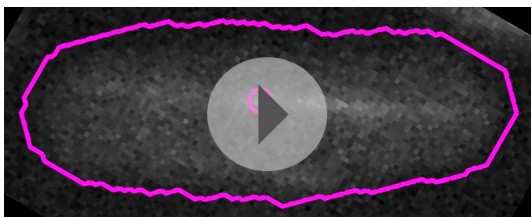

**Video 2**. Summed intensity projections of GFP-tubulin in mitotic fission yeast cell augmented with tracking and segmentation results. The cell outline is shown in magenta. The magenta circles represent tracking of the SPBs from the cut12-tdTomato channel (not shown). These are used to define the green box, which is used to compute spindle intensity. Tracking of the spindle intensity is ceased after the spindle elongates beyond 9 μm. At later stages of elongation more pronounced buckling of the spindle is observed, and microtubules begin to be nucleated in the vicinity of the cytokinetic ring.

cdc25.22 mutation (**Ding et al., 1993**). Interestingly, cells from the cdc25.22 genetic background are enlarged upon entry into mitosis (**West et al., 2001**), and can be compared with the wild-type spindle to provide insight into how the spindle's architecture is altered by increases in cytoplasmic volume.

The earlier spindle reconstructions of spindles in cdc25.22 cells were performed using serial-section electron microscopy (**Ding et al., 1993**), which involves sectioning fixed cells in a particular orientation, imaging each section using transmission electron microscopy and then recording the transverse microtubule positions (**Soto et al., 1994**). These two-dimensional slices are then registered and used to approximate the full three-dimensional object. The main disadvantage of this technique is that movements of the microtubules perpendicular to the imaging plane cannot be fully accounted for, which can cause straightening of microtubules during the registration step. The technique provides accurate estimates of microtubule length and transverse organisation in linear microtubule bundles, but it may be impossible to track strongly curved microtubules, which, by definition, have an orientation that varies along their length. These difficulties are largely overcome by electron tomography where the sample is imaged at numerous orientations (i.e. a tilt series) with respect to the imaging plane (**Soto et al., 1994**).

An interesting feature of the tomographic spindle reconstructions of wild-type cells is the curvature that is present in all of the anaphase B spindles (**Figure 1B**). These deflections are more pronounced in the early anaphase B spindles, where they appear to be inconsistent with the linear spindle morphology that can be observed in cells via light microscopy (**Mallavarapu et al., 1999**; **Tolić-Nørrelykke, Sacconi, et al., 2004b**;

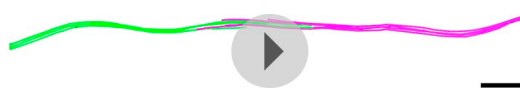

**Video 3**. Electron Tomogram (ET) reconstruction of short anaphase B spindle from **Figure 1B**, top. Scale bar = 0.5 µm.

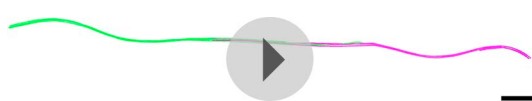

**Video 4**. Electron Tomogram (ET) reconstruction of intermediate length anaphase B spindle from **Figure 1B**, middle. Scale bar = 0.5 µm.

**Video 5**. Electron Tomogram (ET) reconstruction of long anaphase B spindle from **Figure 1B**, bottom. Scale bar = 0.5 µm.

**Khodjakov et al., 2004**). This suggests that the deflections are caused partially or entirely by the standard preparation methods for electron tomography (**Giddings et al., 2001**; **Höög and Antony, 2007**; **Buser, 2010**), and that native spindles have a straighter morphology. In the absence of further confirmation of the degree of microtubule curvature in live cells; we developed the IFTA algorithm to trace the path of the spindle in three dimensions. This enabled us to computationally straighten the spindle, and to infer the microtubule arrangements in the plane perpendicular to the spindle's main axis, independently of any spindle curvature. Together with the microtubule lengths, this corrected information was used in the subsequent analyses.

The organisation of the spindle in wild-type cells resembles the serial-section reconstructions of cdc25.22 fission yeast spindles in many respects (**Ding et al., 1993**), but there are also several notable differences. In both genetic backgrounds, the spindle is highly symmetric from late metaphase onwards with an almost identical number of microtubules nucleated from each spindle pole (**Figure 1C**). The total length of tubulin polymerised in spindle microtubules also appears to remain constant throughout anaphase B, in both conditions, which is accommodated by a gradual reduction in microtubule numbers. The most striking difference is that cdc25.22 spindles contain around twice the quantity of polymerised tubulin. The likely cause for this discrepancy is the cdc25.22 mutation, which causes the allele of the mitotic phosphatase cdc25p to be inactivated by high temperature (**Russell and Nurse, 1986**). The cdc25.22 allele is hypomorphic under the permissive temperatures used by Ding et al., and consequently cells divide at lengths that are around twice that of wild-type cells (**Hagan et al., 1990**). This suggests that the increased quantity of polymerised tubulin in cdc25.22 mutant cells arises from the increased cytoplasmic volume and the corresponding twofold increase in the abundance of tubulin and other spindle assembly factors.

Other differences between the wild-type and cdc25.22 spindles are associated with the transverse organisation of microtubules. In particular, we have observed that, in wild-type cells, the square-packed organisation with a regular 40 nm centre-to-centre microtubule separation is established during metaphase whereas spindles from the same stage in the cdc25.22 mutants appear to have a less ordered midzone architecture (**Ding et al., 1993**). Square packing is present at the midzone of anaphase B spindles in both genetic backgrounds, but the tight hexagonal packing nearer the poles of anaphase B spindles was not observed in cdc25.22 mutant cells, where a more variable transverse architecture was present. These two differences could be explained by the spindle morphology being perturbed by the increased microtubule polymer in cdc25.22 fission yeast cells or the technical differences in the electron microscopy.

## Effects of spindle architecture on its response to compressive forces

The ordered wild-type spindle architecture, the well-defined forces to which it is subjected (**Tolić-Nørrelykke, Sacconi, et al., 2004b**; **Khodjakov et al., 2004**) and the limited quantity of tubulin subunits that appear to be available for its assembly prompted us to investigate the mechanical properties of the complex assembly of microtubules that form the spindle. The mechanical response of slender elastic beams under compression can be described using the Euler-Bernoulli beam theory (**Landau and Lifshitz, 1986**), which states that a beam will remain straight if subjected to forces below a certain threshold but will buckle if the critical force is exceeded. The critical force depends on the length of the beam, the elastic properties of the constituent material (specifically, its Young's modulus, E) and

**Table 1.** Measurements for calculation of critical and absolute tubulin concentration in fission yeast

**Cell dimensions**

| | | |
|---|---|---|
| Radius | $R_c = 1.6 \pm 0.1$ µm | Maximal cell diameter/2 (**Foethke et al., 2009**) |
| Length | $L_c = 14.3 \pm 0.9$ µm | Distance between cell tips at mitotic entry (**Martin and Berthelot-Grosjean, 2009**; **Moseley et al., 2009**) |
| Volume | $V_c = 2\pi R_c3.(L_c/2R_c-1/3)$ $V_c = 106.4 \pm 13.4$ µm$^3$ | Assumes cell is sphero-cylindrical |
| **Spindle properties** | | |
| Steady-state polymer | $27.1 \pm 4.2$ µm | Sum of all the microtubule's length. This study |
| α/β-tubulin heterodimers | $(4.4 \pm 0.6) \times 10^4$ dimers | Each MT has 13-protofilaments with a length of 8 nm (**Howard, 2001**) |
| Relative intensity | $0.16 \pm 0.02$ | Mean fluorescent intensity ratio of spindles longer than 4 µm |
| **Tubulin dimer concentration** | | |
| Abundance | $(20.0 \pm 4.0) \times 10^4$ | Number of dimers in the cell |
| Free pool | $3.61 \pm 0.68$ µM | Corresponds to the critical concentration of MT assembly |
| Polymerized pool | $0.68 \pm 0.13$ µM | |
| Total concentration | $4.30 \pm 0.81$ µM | |

the beam's cross-section (**Figure 2**). The Young's modulus is a measure of the stiffness of a particular material. Intuitively, it can be thought of as the constant that relates the degree of contraction (or strain) in a block of material to the pressure (or stress) exerted on its ends.

The influence of the cross-section on the beam's response to simple loads can be summarised by two orthogonal vectors associated with the scalar values $I_{min}$ and $I_{max}$ (**Figure 2A–D**). These *principal axes* define the direction of the beam's maximal and minimal resistance to bending forces. In civil engineering, anisotropic structures such as the I-beam are typically used as horizontal supports with the stiffer axis oriented in the same direction as the major load to which the beam is subjected (**Gere and Goodno, 2012**). However, under purely compressive forces beams with a constant cross-section will typically buckle in the direction of the most compliant principal axis (**Figure 2E**), and thus the minimal transverse stiffness, $I_{min}$, determines the critical force for beams of this type. It is for these reasons that the columns used as vertical supports in buildings, and perhaps also microtubules themselves, have rotational symmetry and an isotropic bending resistance (**Howard, 2001**; **Gere and Goodno, 2012**).

We first considered how the transverse stiffness increases with microtubule number for idealised hexagonal and square arrays (**Figure 2F** and **Figure 3A,B**). In these models, we assume that microtubules are hollow, elastic cylinders, and that cross-linkers form fixed attachments to the microtubule surface (**Figure 3A**). We initially assume that the cross-linkers contribute a negligible density to the area moment of inertia tensor (w = 0 in **Figure 3A**) (**Claessens et al., 2006**). All stiffness estimates are normalised to the flexural rigidity of a single, 13-protofilament microtubule.

This analysis shows that the beam's minimal transverse stiffness contains peaks corresponding to specific microtubule numbers and organisations (**Figure 3C,D**). These structural motifs have far higher minimal transverse stiffness than arrays containing one fewer microtubule, and almost identical minimal stiffness to arrays that contain an additional microtubule. For example, removing a single microtubule from the 2 × 2 square motif in **Figure 3D** reduces the minimal transverse stiffness by ~1100% while the addition of a further microtubule only gives rise to an increase of ~3%. These large increases in the minimal stiffness correspond to mechanically isotropic organisations, where $I_{min} = I_{max}$.

Although the cellular abundances of midzone proteins, such as ase1p and klp9p, are ~30-fold lower than the tubulin isoforms (**Marguerat et al., 2012**), these proteins have a high molecular weight and a very high local concentration at the spindle midzone in anaphase B (**Fu et al., 2009**). We modelled the effects of these proteins by assuming they have identical material properties to microtubules and form dense bridges in the regions of overlap. The uncertainty in the cross-linker density is then modelled by varying the width, w, of the connections between microtubules (**Figure 3A**). This analysis confirms that peaks in the minimal transverse stiffness are present irrespective of the cross-linker

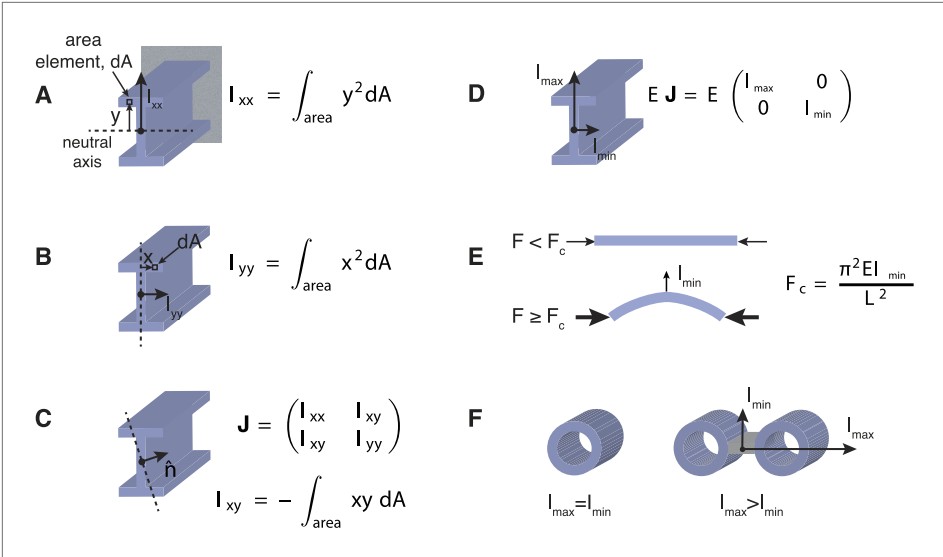

**Figure 2**. Effects of Transverse Organisation on the Critical Force of Prismatic Beams. (**A**) The I-beam is a structural element that is commonly used in civil engineering. If the beam is clamped in a horizontal position at the end furthest from view (grey rectangle), and a transverse force is applied at the other end (in this case upwards), then the beam's resistance is defined by the scalar transverse stiffness, $EI_{xx}$, which is the product of the Young's modulus, $E$, and the area moment of inertia, $I_{xx}$. The area moment of inertia can be computed by dividing the cross-section into area elements, $dA$, and multiplying each area by the square of their distance, $y$, from the neutral axis (dotted line). A sum or integration of this quantity can then be used to compute the area moment of inertia, $I_{xx}$. The neutral axis is perpendicular to the applied force and passes through the centre-of-mass of the beam's cross-section. Intuitively, an increased stiffness can be achieved by placing the beam's material as far from the neutral axis as possible. This property is the rationale behind the design of the I-beam, which typically bears loads in the direction given by $I_{xx}$. (**B**) A similar calculation to **A** can be used to calculate the beam's stiffness in an orthogonal direction. It is usually the case that $I_{xx} > I_{yy}$ for I-beam designs. (**C**) The generalised response of a beam to forces in an arbitrary direction, denoted by the unit vector $\underline{n}$, can be represented by the area moment of inertia tensor, **J**, with entries $I_{xx}$, $I_{yy}$, $I_{xy}$. The tensor, **J**, is a matrix quantity that relates the direction of the applied force to the beam's deflection (*Landau et al., 1986*). For a general tensor matrix, there exist two orthogonal vectors (known as eigenvectors) that represent the directions of the beam's maximal and minimal stiffness. (**D**) The eigenvectors or principal axes of the I-beam point along the x and y-axes. (**E**) Under purely compressive forces a prismatic beam with length, $L$, will buckle in the direction of the most compliant principal axis, which defines the critical force, $F_c$, for beams of this type. (**F**) Beams with a mechanically isotropic transverse organisation, such as microtubules, have a scalar stiffness tensor ($I_{min} = I_{max}$) and degenerate principal axes. The ratio $I_{max}/I_{min} \geq 1$ can be used to quantify the beam's degree of anisotropy. The ratio is one for mechanically isotropic structures such as a single microtubule or a bundle of 4 microtubules arranged in a 2 × 2 square motif.

density, and are therefore associated with the geometry of square and hexagonal arrays (*Figure 3—figure supplement 1*). The structural motifs, such as the triangular motif, the 2 × 2 square motif and the 3 × 3 square motif that are observed in late metaphase and early anaphase B, thus appear to represent features that increase the spindle's stiffness and the force that it can tolerate before buckling.

A notable feature of the stiffness of the idealised arrays, shown in *Figure 3C,D, is* that the mean stiffness increases more rapidly for the square-packed than for the hexagonal bundles. This leads, for example, to four hexagonally-packed microtubules having a mean stiffness of ~21$EI_{MT}$ compared with the 2 × 2 motif's stiffness of ~34$EI_{MT}$. This effect has two causes with the first being the intrinsically higher hexagonal packing density, which is $2/\sqrt{3}$ times greater than equivalent square packing. The second cause is the wider bridging distance between cross-linked microtubules at the midzone compared with the polar regions of the spindle (*Figure 1E,F*). Both of these effects tend to increase the separation between each microtubule centre and the bundle's neutral axis and thus increase the transverse stiffness of the midzone. Since, the bridging distance between microtubules is set by the span of the cross-linking molecules, selection for increased bundle stiffness could be one of the

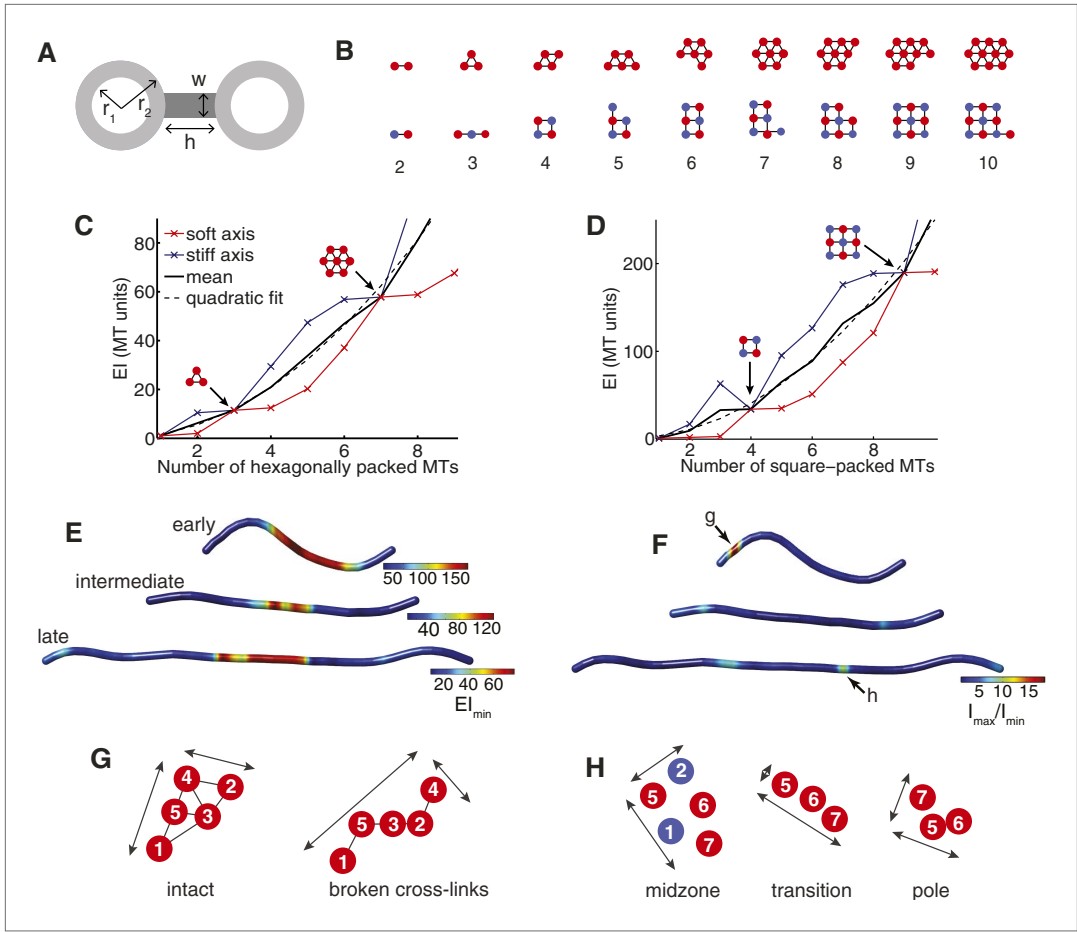

**Figure 3**. The Cross-Sectional Microtubule Organisation Enhances the Spindle's Transverse Stiffness. (**A**) Schematic representation of a pair of bundled microtubules. Microtubules are modelled as hollow cylinders that are linked by rectangular support elements with identical material properties. In all moment of inertia calculations we used $r_1 = 7.5$ nm and $r_2 = 12.5$ nm. (**B**) Transverse organisation of idealised hexagonal and square-packed arrays of microtubules as additional fibres are added. (**C, D**) Eigenvalues of the stiffness tensor for idealised hexagonal and square-packed microtubule arrays. Results are shown for centre-to-centre separations (equal to $h+2\,r_2$) that match those obtained from ET reconstructions. These are 30 nm and 40 nm for hexagonal (polar) and square (midzone) arrays, respectively. The blue and red curves show the small and large eigenvalues of the stiffness tensor. Black arrows represent microtubule organisations with degenerate eigenvalues and isotropic stiffness. The solid black curves show the mean of the large and small eigenvalues, and the dashed black curves a quadratic fit to the mean stiffness. The y-axis (labelled as MT units) shows the stiffness of the bundle divided by the stiffness of one microtubule. (**E**) Minimal transverse stiffness, $EI_{min}$, for three anaphase B spindles with identical units to **C, D**. (**F**) Stiffness anisotropy ratio, $I_{max}/I_{min}$, color-coded over the ET spindle reconstructions. Arrows mark the positions of the regions shown in **G** and **H**. (**G**) An apparent loss of cross-linker integrity at the pole of the spindle is a cause of stiffness anisotropy. Microtubules are drawn to scale, and are numbered consistently between slices. Arrows show the approximate extent of the bundle in two orthogonal directions. The two sections are separated by a distance of 240 nm along the spindle axis. (**H**) Transverse sections showing stiffness anisotropy at the transition from the square-packed crystalline phase to hexagonal phase. Each slice is separated by an axial distance of 200 nm.
The following figure supplement is available for figure 3:

**Figure supplement 1**. Transverse Stiffness of Cross-linked Microtubule Bundles.

explanations for the relatively large size of the major classes of microtubule motors within the spindle (*Carter et al., 2011*; *Scholey et al., 2014*).

We next applied the transverse stiffness tensor calculations (with w = 0) to the cross-sections of ET-reconstructed fission yeast spindles. This calculation can be used to estimate the minimal transverse

stiffness (*Figure 3E*), and the anisotropy ratio, $I_{max}/I_{min}$ (*Figure 3F*) of the three exemplary spindles marked with arrows in *Figure 1C*. These maps illustrate the decrease in the transverse stiffness that is associated with elongation of the anaphase B spindle. This effect is caused by the conservation of polymer mass, which is maintained by completely depolymerising a sub-set of the interpolar microtubules from the elongating spindle. This reduction in microtubule numbers, in turn, leads to the decrease in the spindle's transverse stiffness (*Figure 3C,D*). We also observed that the square-packed architecture, larger number of microtubules and increased bridging distance between cross-linked microtubules at the spindle midzone result in a greater transverse stiffness than in the polar regions for all phases of spindle elongation (*Figure 3E*).

The estimates of the spindle's transverse anisotropy ratio reveal that all of the anaphase B spindles have low mechanical anisotropy throughout their extent apart from localised stretches (*Figure 3F*). These regions may be associated with local loss of cross-linker integrity in early anaphase B spindles, where recruitment of anaphase-specific microtubule bundling factors is likely to be incomplete, and at the transitions from square to hexagonal microtubule packing arrangements in later spindles (*Figure 3G,H*). The narrow width of these transitional regions (also known as phase boundaries) thus appears to enable fission yeast spindles to establish mechanically isotropic microtubule organisations, throughout most of their extent.

Remarkably, inspection of anaphase B spindles from cdc25.22 mutant fission yeast cells (*Ding et al., 1993*) and budding yeast (*Winey et al., 1995*) indicate that they also contain similar structural microtubule motifs. The enlarged cytoplasmic volumes and greater polymer mass in cdc25.22 cells (*Ding et al., 1993*) leads to spindles that retain the nine microtubules at the spindle midzone in a 3 × 3 configuration late into anaphase B. Conversely, late anaphase B spindles in budding yeast contain around 20 μm of polymer, and are often arranged with two long microtubules emanating from each pole (*Winey et al., 1995*), and a 2 × 2 square-packed array in the central spindle. This suggests that the mechanisms of spindle assembly that give rise to mechanically isotropic arrangements are conserved in other yeast species, and are adaptable to the increased abundance of tubulin in enlarged fission yeast cells.

## Estimating the Effective Stiffness of the spindle and its resistance to primary-mode buckling

As the spindle's cross-sections have striking geometric properties that appear to enhance its mechanical properties, an open question concerns whether the length and number of spindle microtubules also increase the critical force that the structure can tolerate before buckling. Since the Euler-Bernoulli beam theory only applies to prismatic beams with a constant transverse organisation, it is also unclear whether the local design rules for microtubule packing endow the non-prismatic spindles with high critical forces (*Landau et al., 1986*; *Gere and Goodno, 2012*). To address these questions, we constructed a simplified computational model of each anaphase B spindle using the Cytosim simulation software (*Nedelec and Foethke, 2007*). The model spindles were then subjected to increasing compressive loads to induce buckling, which then enabled us to calculate the effective stiffness $EI_{eff}$ of the structure, using the relation $EI_{eff} = F_c(L_s/\pi)^2$, where $L_s$ is the spindle length. In each simulation, the computational model incorporates the microtubule lengths determined by electron microscopy and many of *S. pombe* spindle's known biophysical properties (*Table 2*).

In most eukaryotes, the anaphase midzone is defined by the localisation of the anti-parallel bundling protein, ase1p (*Glotzer, 2009*), which binds strongly to the spindle at anaphase onset and occupies a region at the spindle's centre. In fission yeast, the midzone has a width that is approximately constant throughout anaphase B (*Loïodice et al., 2005*; *Yamashita et al., 2005*; *Janson et al., 2007*), which was reproduced in the simulations of each spindle by confining cross-linkers with specificity for anti-parallel microtubules to a cylindrical region at the spindle centre with the same constant width, $L_m = 2.5$ μm. For simplicity, the model does not include binding of cross-linkers to regions flanking the midzone.

After initialising a bipolar spindle with the correct gross organisation (*Figure 4A,B*; *Video 6*), the rate with which cross-linkers detach from the microtubule lattice was set to zero to approximate the slow turnover of midzone proteins (*Schuyler et al., 2003*; *Loïodice et al., 2005*; *Fu et al., 2009*). The poles were then subjected to a linearly increasing force to probe the spindle's elastic response. Simulation results for the longest fission yeast spindle, after subjecting it to forces that increase at different rates, are shown in *Figure 4C*. These numerical experiments indicate that the simulated

**Table 2.** Physical and numerical constants for simulations of spindle stiffness

| Description | Value | Notes |
|---|---|---|
| **Global** | | |
| $k_BT$ | 0.0042 pN.µm | Thermal energy at T = 27°C |
| Viscosity | 1 pN pN.s.µm$^{-2}$ | Viscosity of fission yeast cytoplasm (*Tolić-Nørrelykke, Munteanu, et al., 2004a*) |
| Time step | 0.001 s | Simulations with smaller time-steps produce similar results |
| Microtubules | | |
| Segmentation | 0.1 µm | |
| Flexural rigidity | 20 pN.µm$^2$ | This is EI$_{MT}$ (*Gittes et al., 1993*) |
| Steric radius | 30 nm | Microtubule outer radius + Debye length |
| Steric stiffness | 200 pN.µm$^{-1}$ | per microtubule segment |
| Midzone width | 2.5 µm | (*Loïodice et al., 2005*; *Yamashita et al., 2005*) |
| Spindle Pole Bodies | | |
| Radius | 60 nm | Observed from ET |
| Depth | 100 nm | Observed from ET |
| Stiffness 1 | 1000 pN.µm$^{-1}$ | Appropriate for (*Khodjakov et al., 2004*) (*Tolić-Nørrelykke, Sacconi, et al., 2004b*) (*Toya et al., 2007*) |
| Stiffness 2 | 20 pN.µm$^{-1}$ | Appropriate for (*Kalinina et al., 2013*) |
| Cross-linkers | | |
| Number | 300 | Less than abundance of ase1p (~900 dimers/cell) and klp9p (~1300 dimers/cell) (*Marguerat et al., 2012*). Larger numbers do not alter simulation results |
| Bridging length | 50 nm | Approximate centre-to-centre distance of microtubules bundled by Map65 proteins (*Subramanian et al., 2010*). |
| Link stiffness | 1000 pN.µm$^{-1}$ | Force is Hookean with a non-zero resting length (*Howard, 2001*) |

spindle withstands the increasing force until a threshold is reached and buckling occurs. All of the stochastic simulations were initialised in the same state and behaved identically until a force is applied. However, there are substantial differences in the peak force that can be sustained before the spindle buckles. This transient effect is a known signature of the buckling instability, and becomes more pronounced as spindles are compressed more rapidly. It is for this reason that the estimates of the critical force of the spindle architectures were determined by averaging the force in the final 50 s of the simulation when the system is in equilibrium. The responses from computational models of wild-type spindles, compressed at the same rate (*Figure 4D*), shows that the critical force decreases with spindle length.

A comparison of the critical forces for all of the simulations of ET-reconstructed spindles shows that it scales as $F_c \propto L_s^{-4}$ with spindle length (*Figure 4E*). This fourth-order dependence has a contribution from the beam theory, $F_c \propto EI_{eff}L_s^{-2}$, and another contribution from the conserved polymer mass (*Figure 1H*). The polymer mass conservation gives rise to a quadratic decay in the spindles' effective transverse stiffness ($EI_{eff} \propto L_s^{-2}$, see 'Materials and Methods'). This adverse scaling law suggests that the combination of a pushing mode of force generation with recycling of tubulin subunits and telescopic extension acts to severely limit the forces that can be generated by a spindle architecture of this type.

We next considered whether the number and length of microtubules within each spindle maximise the spindles' critical force. This was addressed by constructing a model to sample alternative spindle architectures with the same polymer mass as wild-type spindles. In this model, the number of microtubules nucleated at each pole is sampled from a stochastic process and the total polymer is distributed randomly between the nucleated microtubules. This comparison reveals that the wild-type fission yeast spindles are substantially stronger than the random model (*Figure 4F*); a property that also applies to the cdc25.22 and the majority of budding yeast spindles (*Figure 4—figure supplement 1*). This analysis also indicates that alternative spindle architectures with a larger critical force occur infrequently under this null model, and suggests that the lengths of wild-type microtubules are regulated to increase the spindle's effective stiffness.

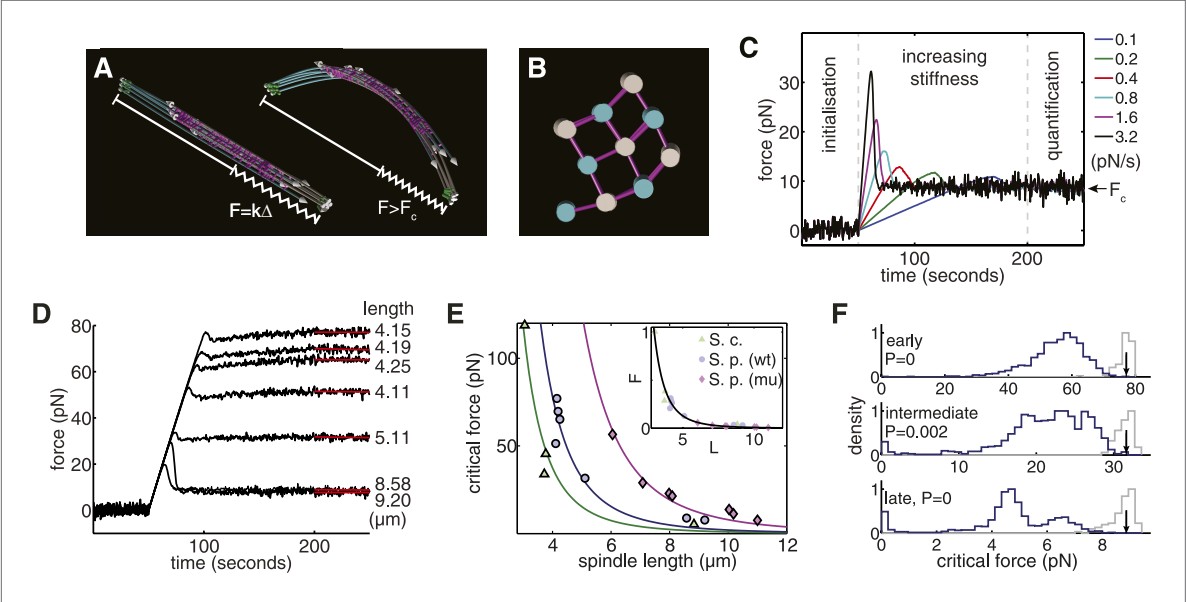

**Figure 4**. Computational Reconstructions of the Spindle can be used to Estimate its Effective Stiffness. (**A**) Computational reconstruction of an early anaphase B spindle before and after the critical force is exceeded. The spindles are subjected to compression by attaching a spring between the spindle poles. The elastic constant is then increased to probe the spindle's response to force. (**B**) Stochastic initialisation conditions can reproduce the 3 × 3 organisation at the midzone of a spindle in early anaphase **B**. Microtubules are depicted with a diameter of 25 nm. (**C**) Model response of the longest fission yeast spindle to forces that increase at different rates. The SPBs are held at the spindles' contour separation for the first 50 s of the simulation, when cross-linkers are allowed to form attachments between the two halves of the bipolar spindle. At t = 50 s, the elastic constant of the spring connecting the two SPBs is then set to zero and its resting length reduced to Δ = 1 μm less than the SPB's resting separation. The elastic constant is then increased linearly over the subsequent 150 s of the simulation to exert progressively larger force on the spindle. Each colour, represented in the legend, denotes a different force increase in units of pNs$^{-1}$. The spindles bear the increasing compressive loads until a threshold is reached and the force decays before plateauing at the equilibrium (critical) force, $F_c$. In the final 50 s of the simulation (marked quantification), the spring constant of the elastic element connecting the SPBs is maintained at its maximal value, and the critical force quantified by averaging the force-response profile. (**D**) Response of fission yeast spindle models to compressive forces. The contour length of each spindle is noted in the right-hand column. Curves show the median critical force from N = 100 stochastic simulations. (**E**) Dependence of critical force on spindle length for models of w.t. *S. pombe*, cdc25.22 *S. pombe* and *Saccharomyces cerevisiae* anaphase B spindles. Each point indicates the median critical force calculated from N = 100 simulations. Curves show $F_c = AL_s^{-4}$ fits to the simulation results, where the only fit parameter is the pre-factor, A. Inset shows normalised force $F_c/A = L_s^{-4}$ for all spindle types. This rescaling highlights the universality of the relationship between spindle length and critical force. (**F**) Comparison of the critical force of fission yeast spindles (grey histograms, N = 100) with a null statistical model (blue histograms, N = 10$^3$). p-values refer to the probability that a random spindle has a critical force greater than the median wild-type spindle.

The following figure supplement is available for figure 4:

**Figure supplement 1**. Compressive Strength and Optimality of Yeast Spindles.

## The number of interpolar microtubules is regulated precisely in the anaphase B spindle

To gain insight into the origin of the high critical force of wild-type spindles, we investigated alternative architectures that are expected to have the highest resistance to compressive forces. In spindles where anti-parallel cross-linkers are confined to a region at the spindle centre, microtubules that are too short to reach the midzone cannot form associations with a partner from the opposing SPB (*Figure 5A*). These microtubules make only a small contribution to the stiffness of the structure, and represent an inefficient use of the fixed length of polymerised tubulin, $L_T$, that is available for building the spindle. The span of a microtubule that projects beyond the midzone region also makes an inefficient contribution to the spindle's stiffness. Maximally efficient overlap at the spindle midzone can thus be achieved by ensuring that all microtubules terminate at the furthest edge of the midzone, and have a uniform length, $L_{MT} = (L_s + L_m)/2$, where $L_s$ is the spindle length, and $L_m$ the length of the overlap (*Figure 5B*). The conservation of polymer mass then gives the following equation for the number of microtubules, $N_{MMO}$, that are present in these maximal midzone-overlap (MMO) spindle architectures

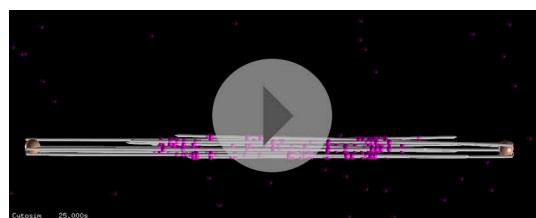

**Video 6**. Simulation of Spindle Subjected to Compressive Forces. In the first 50 s of the simulation, the force exerted on the poles of the spindle is zero, and cross-linkers are allowed to bind and unbind from the microtubule lattice to form attachments between the two halves of the bipolar spindle. After 50 s the force on the spindle increases linearly until buckling is induced (**Figure 4C,D**).

$$N_{MMO} = \frac{2L_T}{L_s + L_m}$$

Simulations probing the force-response of MMO spindles reveal critical forces that are, on average, only 8% higher than those measured for reconstructed wild-type and cdc25.22 spindles (**Figure 5—figure supplement 1**). This suggests that fission yeast cells exert precise global control on microtubule number and length.

A comparison of the observed number of interpolar microtubules in wild-type spindles, $N_{obs}$, with the theoretical prediction from equation 1 (**Figure 5C**, blue curve) confirms that microtubule numbers are under precise control in fission yeast. This curve also indicates that the midzone width and total length of polymerised tubulin in wild-type cells enable the spindle to conform to the first $3 \times 3$ midzone organisation at the start of anaphase B where spindle-derived forces must remove residual catenation between the chromosome arms (**Renshaw et al., 2010**; **Petrova et al., 2013**) and remodel the nuclear envelope (**Yam et al., 2011**). The $2 \times 2$ motif can also be maintained throughout spindle elongation up to lengths of around 10 μm (**Figure 5C**), which coincides with the length at which GFP-tubulin spindles are observed to begin disassembly (**Figure 1A**; **Videos 1, 2**).

Remarkably, the theoretical curve for microtubule number in cdc25.22 spindles containing around twice the mass of polymerised tubulin (**Figure 5C**, red curve), is also in very close agreement with observed number of interpolar microtubules. A direct comparison between $N_{obs}$ and $N_{MMO}$ for wild-type and cdc25.22 spindles (**Figure 5D**) reveals that all but four of the fourteen spindles deviate from the theoretical model by less than a single microtubule, thus indicating precise scaling of microtubule number to changes in spindle length and the availability of polymerised tubulin.

## Elastic reinforcement may increase the forces that the spindle can sustain

Whilst microtubule number and length appear to be regulated precisely in the fission yeast spindle, the critical force of the longer spindles, at around 10 pN (**Figure 4E,F**), is relatively small compared with other intracellular forces. For example, whilst this force exceeds the viscous drag on daughter nuclei in wild-type cells (see 'Materials and methods') it is comparable to the force generated by two kinesin motors (**Gittes et al., 1996**; **Visscher et al., 1999**). The late anaphase B spindle would therefore appear to have an inefficient architecture for generating forces to segregate lagging or concatenated chromosomes (**Pidoux et al., 2000**; **Courtheoux et al., 2009**; **Petrova et al., 2013**). We therefore investigated whether external reinforcement (**Brangwynne et al., 2006**) of the spindle might lead to an increase in the forces that the interpolar microtubules can support under these conditions.

In simulations of the spindle, exceeding the critical forces leads to the spindle being buckled into a shape with a single maximum, which we refer to as primary-mode buckling (**Figure 4A**). Whilst this behaviour is predicted by the Euler-Bernoulli beam theory (**Landau et al., 1986**), the theory also predicts that the beam can buckle with shorter wavelengths. However, since these higher-order modes (i.e. with two or more maxima) occur at higher critical forces (and energy) than buckling in the primary mode, they are expected to relax into the primary mode over longer time scales. It has, however, been shown, using experiments on cells and macroscopic analogues (**Brangwynne et al., 2006**), that laterally reinforcing microtubules with a dense elastic meshwork can lead to stable buckling over much shorter wavelengths. For microtubules subjected to an elastic confinement, the critical force that can be supported is (**Brangwynne et al., 2006**)

$$f_c = 8\pi^2 \frac{EI}{\lambda^2}$$

which is similar in form to the classical Euler-Bernoulli formula but, in this case, the critical force depends on the buckling wavelength, λ, rather than the length of the beam. The buckling wavelength

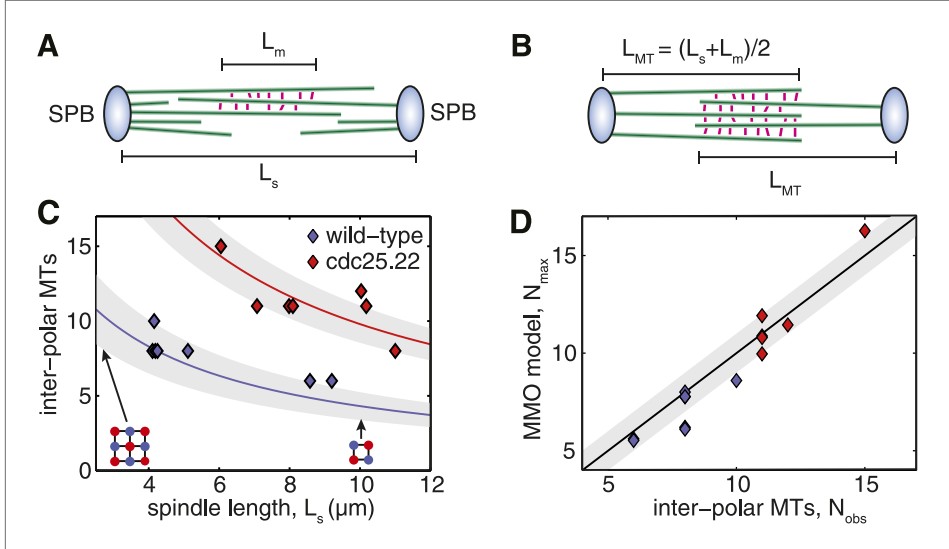

**Figure 5**. Fission Yeast Spindles Scale Microtubule Lengths and Number for Maximal Overlap at the Spindle Midzone. (**A**) Schematic representation of a fission yeast spindle of length $L_s$. Green lines represent microtubules and magenta lines anti-parallel cross-linkers, which bind to the midzone region of width $L_m$. Blue shaded ellipses represent the spindle pole bodies (SPBs). (**B**) Maximal anti-parallel bundling of microtubules at the spindle midzone is achieved if microtubules have a monodisperse length $L_{MT} = (L_s + L_m)/2$. (**C**) Number of interpolar microtubules in wild-type and cdc25.22 mutant fission yeast cells with respect to spindle length (blue and red diamonds, respectively). Trend lines show theoretical calculation of microtubule number in MMO spindle architecture based on the spindle length, $L_s$, and average total polymer, $L_T$. For wild-type cells $L_T$ = 27.1 S.D. 4.2 μm (blue curve with standard deviation represented by grey shaded area) whilst for cdc25.22 cells $L_T$ = 61.2 S.D. 7.8 μm (red curve). (**D**) Comparison of interpolar microtubule number with predictions of the MMO model. The black line shows trend for exact agreement with grey bars representing 1 microtubule deviation.

The following figure supplement is available for figure 5:

**Figure supplement 1**. Critical force of Wild-type and Cdc25.22 Mutant Fission Yeast Cells.

is determined by the transverse stiffness of the fibre and a parameter, α, which measures the strength of the elastic confinement

$$\lambda = 2\pi \left( \frac{EI}{\alpha} \right)^{1/4}$$

We investigated this effect by imposing an elastic confinement on the spindles from early, intermediate and late stages of elongation (*Figure 6A–C*; *Videos 7–9*). As before, the spindles were subjected to increasing compressive loads until they underwent buckling. A visualisation of the shape adopted by the shortest anaphase B (*Figure 6A*) reveals that its profile transitions from buckling in the primary mode when unreinforced (α = 0) to second-order buckling for an elastic confinement α = 40 Pa, with intermediate configurations containing both components (α = 20 Pa). A similar transition occurs for longer spindles (*Figure 6B,C*) but with lower elastic confinement.

A comparison of the force-response curves for all of the simulated wild-type spindle architectures (*Figure 6D*) shows that the critical force that can be withstood increases with greater elastic confinement. This effect is most dramatic for the longest anaphase B spindles where large elastic confinements can increase the critical forces by around two orders of magnitude. This effect is caused by the critical force of laterally reinforced spindles decreasing much more slowly with increasing spindle length (*Figure 6E*). These results are in good agreement with the theory for reinforced elastic rods (*Brangwynne et al., 2006*), which predicts a critical force, $F_c \propto \sqrt{EI}$. In the case of the spindle, this leads to a $F_c \propto L_s^{-1}$ dependence, which is caused by the reduction in the effective transverse stiffness ($EI_{eff} \propto L_s^{-2}$) that occurs as a consequence of spindle elongation. These results suggest that a small degree of

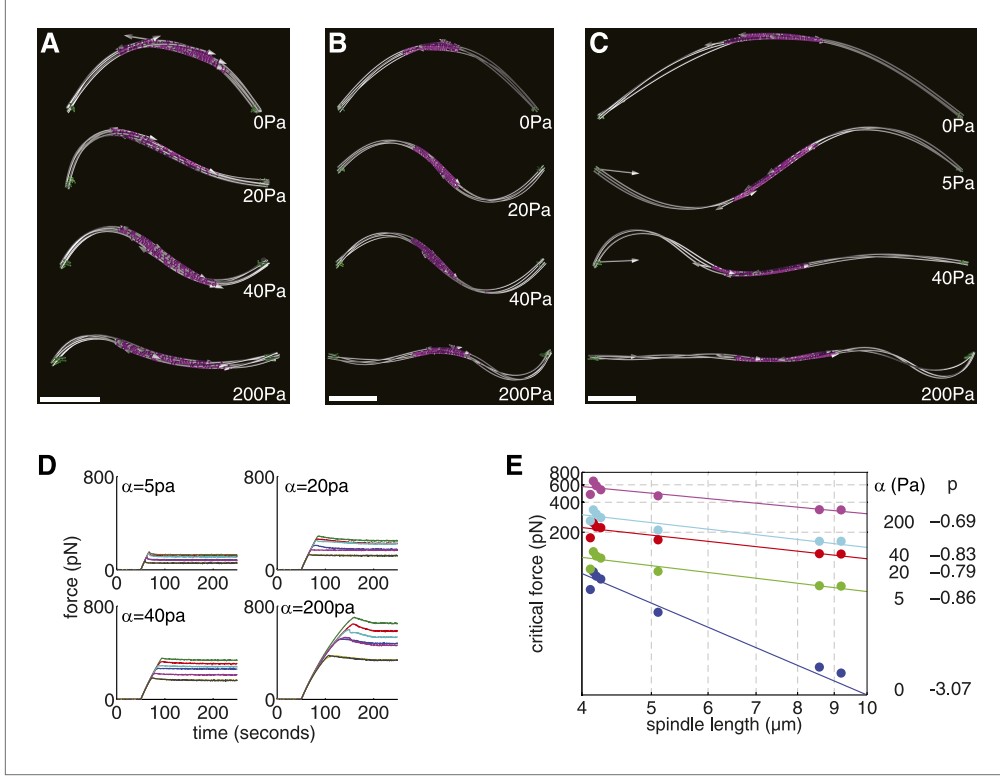

**Figure 6**. Elastic Reinforcement Enhances the Compressive Loads that can be Borne by the Spindle.
(**A**–**C**) Equilibrium shapes adopted by exemplary anaphase B spindles after the critical force is exceeded. The degree of lateral reinforcement, α, is shown alongside each simulation. Scale bars = 1 μm. (**D**) Force-response of reconstructed anaphase B spindles (median critical force calculated from the results of 50 stochastic simulations). (**E**) Relationship between the critical force of laterally reinforced spindles and their length. Lines reflect power law fits $F_c = AL^p$, with exponent, p, shown to the right of each curve.

lateral reinforcement, compared with the ~1 kPa found in interphase mammalian cells (***Brangwynne et al., 2006***), could greatly increase the forces that the anaphase B spindle can sustain. The scaling law relating the critical force to the length of a reinforced spindle could also facilitate its elongation in enlarged fission yeast cells.

## Discussion

The design of structures that maximise strength given certain material constraints is a class of problem that appears commonly in mechanical engineering. This study has shown that yeast cells have evolved a spindle architecture that incorporates similar design principles to those used in man-made machines. These features may have evolved to overcome constraints on the abundance and material properties of proteins that form the spindle.

### Material properties of the spindle

In the case of the yeast spindle, the relatively high abundance of midzone proteins (***Fu et al., 2009***; ***Marguerat et al., 2012***) and the force-resistant, non-slip binding of kinesin molecules to the microtubule lattice (***Gittes et al., 1996***; ***Visscher et al., 1999***) suggest that the spindle is densely cross-linked at the midzone. Furthermore, the dramatic remodelling of the spindle's transverse architecture at the boundary between the midzone and flanking regions of the spindle (***Figure 3G,H***) further suggests that the cross-linkers that bind to the polar regions also form rigid bridges between pairs of microtubules. Finally, the slow but uniform rate of spindle elongation (***Figure 1I***) suggests that the midzone-bound motors operate in a regime that is far from their stall force. In simulations of the anaphase B spindle, introducing a sufficient number of crosslinkers leads to reduced movement of the filaments at the overlap zone, and causes this part of the spindle to behave elastically (***Claessens et al., 2006***).

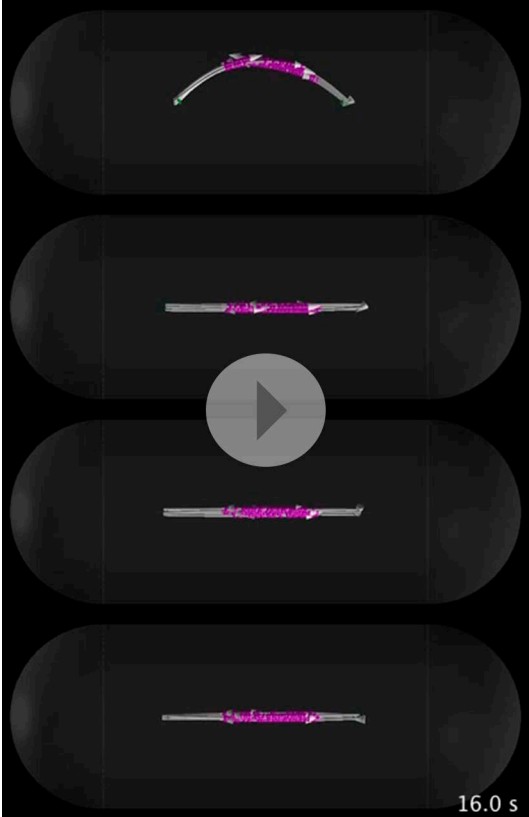

**Video 7**. Simulation of Short, Intermediate and Long Example Spindles Subjected to Compressive Forces with Differing Degrees of Lateral Reinforcement. Scale is set by the size of the enclosing cell, which measures 11 μm in length between the opposite cell tips. From the image at the top downwards, the spindles are subjected to increasing confinement with α = 0, 20, 40, 200 Pa, respectively for the short and intermediate length spindles. For the longest spindle, α = 0, 5, 40, 200 Pa.

An important distinction between the spindle and macroscopic machines is that it must self-organise from local interactions between protein components that are many orders of magnitude smaller in size. This necessitates the use of chemical and physical properties of these proteins to generate a particular microtubule organisation. The key feature that endows the yeast spindle with rigidity is the crystalline microtubule architecture that is present throughout its extent, and which takes the form of hexagonal packing in the polar regions of the spindle and square-packing at the spindle midzone.

The advantage of crystals over more disordered microtubule packing arrangements is that the number of interactions between a microtubule and its neighbours is increased (*Figure 7A*). This increased connectivity enhances the spindle's stiffness by ensuring that transverse rearrangements of microtubules that lead to bundle warping are prevented, and that the structural integrity of the bundle is thus maintained. Arranging the crystalline units in each cross-section with rotational symmetry enhances the minimal transverse stiffness still further. This combination produces stiff microtubule arrays with isotropic responses to force; similarly to the designs for load-bearing columns used in civil engineering since antiquity (*Thompson, 1942*; *Gere and Goodno, 2012*).

Similar bundle design principles have also been observed in horseshoe sperm acrosomes, which are long, finger-like actin projections that enable the sperm cell to penetrate the 30 μm thick jelly coat that surrounds the egg (*Schmid et al., 2004*; *Shin et al., 2004*). These dense, highly cross-linked, crystalline actin bundles are likely to have been selected for resistance to compressive forces (*Schmid et al., 2004*), and the composite struc-

ture has a Young's modulus that is similar to that of the constituent actin filaments (*Shin et al., 2004*). The convergent evolution of a crystalline architecture in cytoskeletal bundles that resist compressive forces may be one factor that has led to the distinct transverse arrangements that are observed in different populations of spindle microtubules (*McDonald et al., 1979*, *1992*; *Mastronarde et al., 1993*) and may reflect the relative contributions of tensile and compressive forces in each system (*Wühr et al., 2009*; *Goshima and Scholey, 2010*).

Two additional features of yeast spindles that also make contributions to their high critical forces are the control of microtubule length and number. We have provided evidence that the regulation of these properties enables the fission yeast spindle to use the available microtubule polymer to form specific square-packed midzone motifs at critical points during mitosis (*Figure 1C*; *Figure 5C*). This property also applies, albeit less precisely, to the budding yeast spindle, where a lower midzone width ($L_m \approx 2$ μm, (*Schuyler et al., 2003*)) enables the reduced quantity of polymerised tubulin ($L_T = 17.9 \pm 6.5$, (*Winey et al., 1995*)) to be used to form $3 \times 3$ square-packed motifs at the onset of anaphase, and to maintain $2 \times 2$ motifs up to a length of around 8 μm. It remains to be seen whether a similar pattern, with coevolution between microtubule polymer mass, midzone width and final spindle extension, is present in other ascomycete yeast species with a similar spindle organisation (*Horio and Oakley, 2005*; *Roca et al., 2010*). It will also be interesting to investigate how motors (*Fu et al., 2009*), cross-linking molecules (*Janson et al., 2007*) and regulators of microtubule stability (*Bratman and Chang, 2007*)

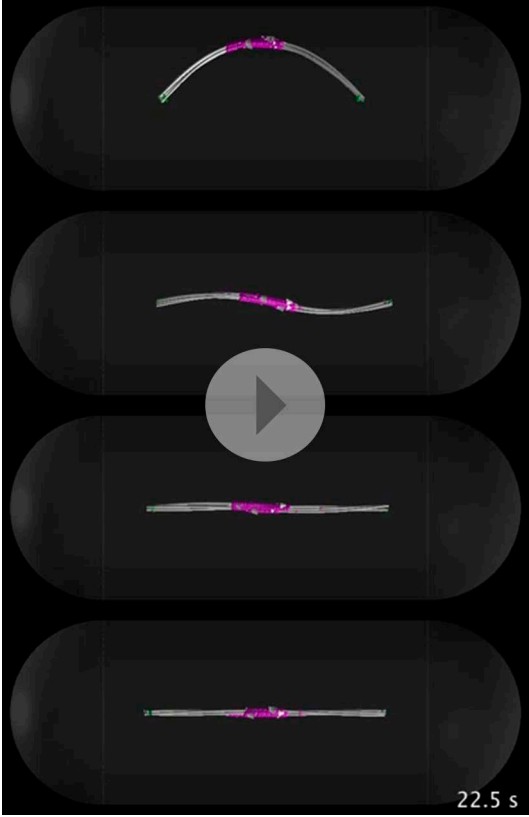

**Video 8**. Simulation of Short, Intermediate and Long Example Spindles Subjected to Compressive Forces with Differing Degrees of Lateral Reinforcement. Scale is set by the size of the enclosing cell, which measures 11 µm in length between the opposite cell tips. From the image at the top downwards, the spindles are subjected to increasing confinement with α = 0, 20, 40, 200 Pa, respectively for the short and intermediate length spindles. For the longest spindle, α = 0, 5, 40, 200 Pa.

collaborate to form cytoskeletal bundles with this remarkable degree of precision.

A surprising feature of the spindle is that the regulation of its architecture is maintained in enlarged cdc25.22 cells. This suggests that the mechanisms of spindle assembly can adapt its morphology to changes in cell size. This adaptive control may be required to enable the fully elongated spindle to reach the cell poles in fission yeast cells of variable size (*Hagan et al., 1990*). This is a property that fission yeast shares with metazoan embryos, which contain spindles that are observed to scale in size with cytoplasmic volume during early development (*Good et al., 2013*; *Hazel et al., 2013*). However, the evolutionary pressure that has led to this property being selected in unicellular yeast cells is less obvious. One possible explanation is that the spindle has evolved to be robust to natural variations in the size of the cell at mitotic entry (*Martin and Berthelot-Grosjean, 2009*; *Moseley et al., 2009*) or noise in the abundance of different spindle assembly factors (*Kaern et al., 2005*). Another possibility is that spindle scaling has been selected to respond to differentiation of yeast cells into the alternate forms that occur during mating or hyphal growth (*Niki, 2014*).

## Spindle design principles and the generation of force

A vital consideration in the design of engineered systems is tolerance, which specifies the range of performance that is required from a component or device. In studies of spermatocyte spindles and related systems (*Nicklas, 1988*), it was found that the maximal force that can be exerted on chromosomes is typically several orders of magnitude larger than that required to overcome viscous drag from the surrounding cytoplasm. The forces that are applied to chromosomes are likely to vary according to the phase of mitosis and the state of the attachments between kinetochores and the spindle. However, direct measurements of these forces have only been possible in a small number of cell types (*Nicklas, 1983*).

In vitro studies of purified budding yeast kinetochores (*Akiyoshi et al., 2010*) found that the tensile force exerted on a single microtubule influences the life-time of its association with the kinetochore. The most long-lived kinetochore-microtubule attachments take place under loads of around 5 pN, with weaker or stronger forces leading to increased rates of detachment (*Akiyoshi et al., 2010*). This behaviour is consistent with the in vivo regulation of kinetochore attachments, where tension is used to detect chromosomes that are properly bi-oriented on the metaphase spindle (*Bouck et al., 2008*; *Tanaka, 2010*). As the positions of the 16 kinetochore pairs in budding yeast are highly consistent during metaphase (*Joglekar et al., 2009*), when microtubule attachments are at their most long-lived (*Tanaka, 2010*), this suggests that the total tensile force on the budding yeast spindle could be around 80 pN at the end of metaphase.

Fission yeast spindles have complex centromeres, which more closely resemble those of higher eukaryotes (*Malik and Henikoff, 2009*), with around three microtubules occupying each kinetochore (*Ding et al., 1993*) compared with the single microtubule in budding yeast (*Tanaka, 2010*). The coordination between microtubules within composite kinetochore-fibres is currently unknown but

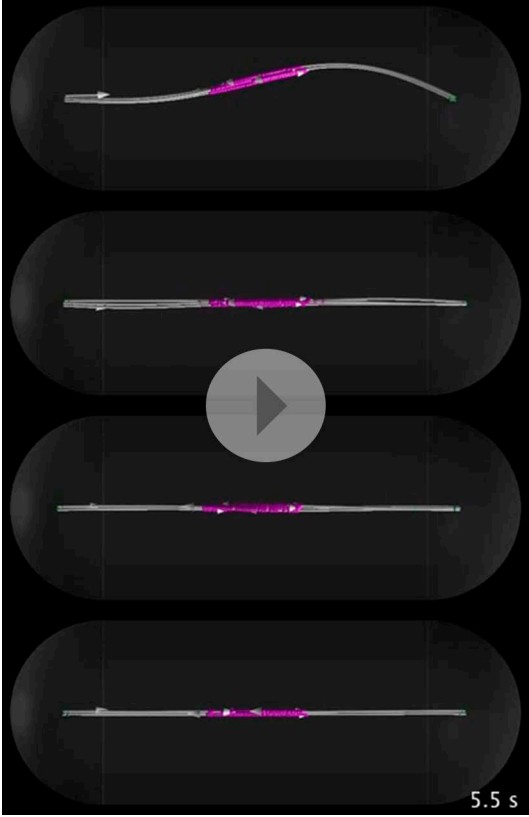

**Video 9**. Simulation of Short, Intermediate and Long Example Spindles Subjected to Compressive Forces with Differing Degrees of Lateral Reinforcement. Scale is set by the size of the enclosing cell, which measures 11 μm in length between the opposite cell tips. From the image at the top downwards, the spindles are subjected to increasing confinement with α = 0, 20, 40, 200 Pa, respectively for the short and intermediate length spindles. For the longest spindle, α = 0, 5, 40, 200 Pa.

extrapolating the in vitro estimates from budding yeast to the three chromosomes in fission yeast would suggest a force in the range of ~50 pN. This value is well below the critical force of unsupported early anaphase spindles, and suggests that shorter, stiffer metaphase spindles (*Figure 1C*) can easily generate the required forces without external mechanical reinforcement.

The forces that are exerted on the anaphase B spindle have a well-defined directionality (*Tolić-Nørrelykke, Sacconi, et al., 2004b*; *Khodjakov et al., 2004*), but their magnitude at the critical early stages of elongation is unknown due to the complex organisation of chromatin around the spindle (*Stephens et al., 2013*) and the need for re-modelling of the nuclear envelope (*Yam et al., 2011*). However, in later anaphase B, after the nuclear envelope splits into a 'dumbbell' morphology, the major external loads opposing spindle elongation are likely to arise from cytoplasmic resistance to the movement of daughter nuclei (*Lim et al., 2007*). A rough estimate for the compressive forces acting on the spindle at this stage of mitosis can thus be estimated by treating the cytoplasm as a viscoelastic fluid.

The viscous drag exerted on the poles of the spindle can be calculated using various properties of the fission yeast cell (*Foethke et al., 2009*)(*Table 3*), whilst the relaxation of nuclei in ablated spindles (*Khodjakov et al., 2004*) can be used to determine the relative magnitude of the viscous and elastic forces (*Figure 7B*). The final estimate for the total resistive force acting on the poles of the elongating spindle is subject to a large degree of uncertainty, but is of the same order of magnitude (~1–10 pN) as the critical force of longer anaphase B spindles. This suggests that the wild-type spindle can support the drag forces that resist its elongation under normal conditions. However, the spindle may be subjected to larger forces if chromosomes are concatenated or are not properly bi-oriented in metaphase (*Pidoux et al., 2000*; *Courtheoux et al., 2009*; *Petrova et al., 2013*).

Under increased loads, the klp9p motors that bind to the spindle midzone (*Visscher et al., 1999*; *Fu et al., 2009*; *Marguerat et al., 2012*) could theoretically generate up to ~$10^3$ pN, while each depolymerising microtubule, of which there are around nine in fission yeast cells during anaphase A (*Ding et al., 1993*), can produce a maximal force of ~40 pN (*Grishchuk et al., 2005*). A possible mechanism to ensure tolerance of the anaphase B spindle against chromosome segregation errors would be to externally reinforce the spindle with an elastic material (*Mitchison et al., 2005*; *Brangwynne et al., 2006*). It remains to be determined whether the anaphase B spindle is reinforced, but candidates for providing this mechanical support have been identified and include actin, which is required for accurate chromosome segregation in fission yeast (*Gachet et al., 2001*; *Meadows and Millar, 2008*), the nuclear envelope (*Lim et al., 2007*; *Yam et al., 2011*) and mitotic chromosomes themselves, which surround the interpolar microtubule axis of yeast spindles (*Stephens et al., 2013*).

Whilst this study has provided evidence that the architecture of the beam-like fission yeast spindle is sculpted for the generation of pushing forces, a more general question concerns why this linear morphology was selected in yeast cells compared with the more conventional ellipsoid spindles observed

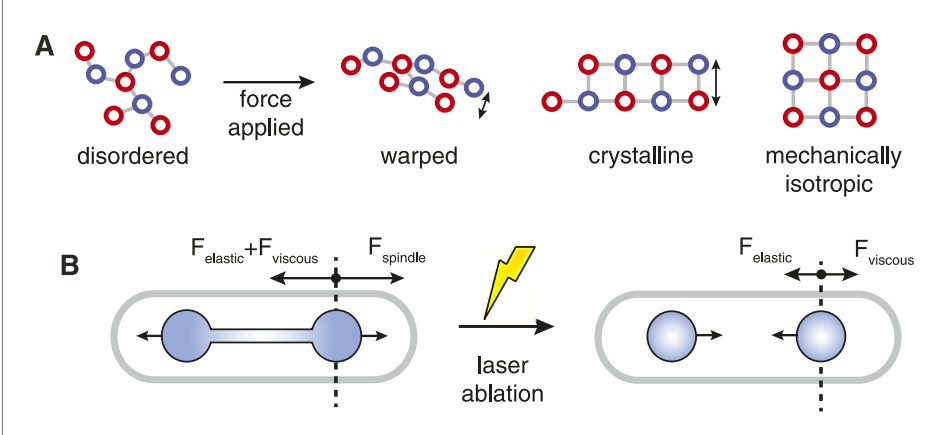

**Figure 7**. Spindle Resistance To Compressive Forces. (**A**) Applying compressive force to a sparsely connected microtubule bundle leads to warping. This increases bundle anisotropy and leads to a reduction in the critical force. In a crystalline microtubule array, microtubules can associate with a larger number of neighbouring microtubules. This constrains microtubule movement and increases the transverse stiffness. Arranging the crystal unit cells with rotational symmetry leads to architectures with an isotropic resistance to bending forces and an increased minimal transverse stiffness. (**B**) In late anaphase B, the forces driving spindle elongation are balanced by the viscoelastic response of the cytoplasm and an elastic force. After ablation of the spindle midzone, this elastic force is opposed by the viscous drag.

in metazoans and many other eukaryotes. One possible functional explanation for the change in spindle morphology is that fungal cells have evolved a stiff polysaccharide cell wall that encloses the plasma membrane (**Bowman and Free, 2006**). This innovation could then have reduced the cell's dependence on the cytoskeleton for providing motility and structural support, and enabled reduced expression of the tubulin and actin isoforms (**Pollard et al., 2000**; **Marguerat et al., 2012**). In yeast cells with a small cytoplasmic volume, the beam-like spindle morphology may have been selected to segregate chromosomes with reduced use of structural components and their associated metabolic cost to the cell. In this respect, the yeast spindle is indeed remarkably efficient, as chromosomes are transported over similar distances to those in many somatic mammalian cell types (**Goshima and Scholey, 2010**), but with a structure that has a 1000-fold smaller volume.

The mitotic spindle displays a diversity of sizes and shapes across different cell types. We have identified structural features of the fission yeast spindle that enable it to use a small quantity of raw material to exert large forces that drive the segregation of chromosomes. This represents a first step towards an understanding of the relationship between the mechanisms of force generation and the architecture of the cell division machinery in eukaryotic cells.

## Materials and methods

### Microscope and yeast culture

Standard *S pombe* genetic and molecular biology techniques, including media, were used as described in the 'Nurse Lab Manual' (**Roque et al., 2010**). Cells were grown and imaged in minimal medium EMM2 supplemented with amino acids where necessary (final concentration of 250 mg/l). For image acquisition, cells were collected and placed in a 35 mm glass-bottom culture dish (MatTek Corp., Ashland, MA, USA) coated with lectin (from *Bandeiraea simplicifolia*, Sigma L2380). Cells were further immobilised with a 2% agarose/EMM2-coated coverslip, which was attached to the glass-bottom dish using VALAP. The partially sealed coverslip was submerged in liquid EMM2 media to prevent drying and deoxygenation of the growth media.

Confocal images were obtained with a Carl Zeiss Axiovert 200 M microscope equipped with a PerkinElmer RS Dual spinning disc system. The Argon Krypton line laser was used at wavelength of 488 nm for GFP signal detection. Images were collected using a 63× oil immersion objective (Plan-Apochromat, NA 1.4) coupled to a Hamamatsu C9100-50 EMCCD camera (Hamamatsu, Japan) with a pixel size of

**Table 3.** Estimates of forces resisting spindle elongation

| Description | Variable | Value | Notes |
|---|---|---|---|
| Cell radius | $r_{cell}$ | 1.6 µm | (**Foethke et al., 2009**) |
| Daughter nuclei radius | $r_{nucleus}$ | 0.8–1.0 µm | This study (ET estimate)−$2^{1/3}$ × radius of interphase nucleus (**Foethke et al., 2009**) |
| Cytoplasmic viscosity | $\eta_{cell}$ | 1 pN.s.µm$^{-2}$ | (**Tolić-Nørrelykke, Munteanu, et al., 2004a**) |
| Spindle pole separation speed | $v_s$ | 0.9 µm.min$^{-1}$ | This study |
| Drag coefficient | $\gamma$ | 25-110 pN.µm$^{-1}$.s | See 'Materials and methods' |
| Relaxation time | $\tau = \gamma/k$ | 9.7 s | Exponential fit to collapse of long spindle (**Khodjakov et al., 2004**), *Figure 7B* |
| Equilibrium position | $x_0$ | 1.4 µm | Second fit parameter |
| Drag force | $F_v = \gamma.v_s/2$ | 0.2–0.8 pN | |
| Elastic force | $F_e = x_0.\gamma/\tau$ | 3–16 pN | This force is dramatically decreased after the nucleus adopts a dumbbell configuration |
| Total force | $F_e + F_v$ | 4–17 pN | |

8 µm. The microscope was controlled with the Ultraview acquisition software (Perkin Elmer, Foster City, CA US). All experiments were performed inside a climate control box (EMBL, Heidelberg, Germany) at a constant temperature of 28°C. Live-cell imaging of fission yeast spindles was carried out using spinning disk confocal microscopy from 15 focal planes separated by a distance of 0.5 µm, a set-up similar to that used to quantify the cellular abundance of proteins localised to the cytokinetic ring in fission yeast (**Wu and Pollard, 2005**).

## Segmentation and tracking of live-cell images

A maximum likelihood image deconvolution algorithm was applied to the images to improve the signal-to-noise ratio of both the tubulin fluorescent marker (**Bratman and Chang, 2007**) and the spindle pole body (SPB) marked with cut12-tdTomato (**Samejima et al., 2010**). After deconvolution, maximum-intensity projections of the SPB marker and summed-projections of the GFP-tubulin marker were used to track the poles of the spindle and to segment the cytoplasmic volume, respectively, but were not used to quantify the intensity of either the spindle or cytoplasm. The segmentation of the cells was performed using a three-dimensional region-growing algorithm (**Gonzalez, 2010**). A seed-point in the cellular background was selected and propagated to the entire image stack, using a threshold of 30% of the mean region intensity as the voxel inclusion criterion. Erosion and dilation operations were then used to remove isolated bright pixels and smooth the cellular outlines. Apart from deconvolution, all analyses of images were carried out using Matlab (The MathWorks inc.). Scripts for performing image analysis are provided in *Source Code 1*.

The tracking of SPB positions was performed using the µ-Track image analysis package (**Jaqaman et al., 2008**). The splitting of a single SPB into two distinct and persistent tracks was used to identify mitotic events in a field of asynchronously dividing cells. Each mitotic event was then mapped to a single segmented cell that was used in subsequent analyses. The background signal was estimated by fitting a histogram to the pixel intensities in each field of view. The background signal was subtracted from the pixel intensity measurements to obtain an absolute estimate of fluorescent intensity. For each cell in mitosis, the background-corrected pixel intensity was integrated over the cytoplasmic volume to obtain a measurement of the total fluorescent intensity within the cell. Large reductions in the integrated cellular intensity were used to diagnose cells that moved significantly during the acquisition period. All acquired cells were also checked by manual inspection of the images augmented with the segmentation and tracking results.

The intensity of GFP-labelled spindle was determined by considering a rectangular window centred on the spindle with a length defined by the positions of the two SPBs. The width of the window was 1.02 microns (8 pixels), which is sufficient to collect the light diffracted by the objective lens (λ ≈ 200 nm). Increasing the width of the window to as much as 1.53 microns did not lead to a qualitative change in the results. Pixel intensity values within the window were determined by linear interpolation to account

for rotation and translation of the pixel positions with respect to the sensor array. These intensity values were corrected for the local cytoplasmic fluorescent intensity, and summed to obtain an estimate of the fraction of the cellular fluorescence that is associated with spindle microtubules.

In order to calibrate the spindle intensities to the tubulin polymerised in microtubules, the polymer within each ET spindle was compared to the intensities of the mitotic cells where pole-to-pole length matches that of the ET spindle most closely (*Figure 1—figure supplement 1*). This relationship was fit with a linear function a.x + b. The small value of b, and the relatively high coefficient of determination support a linear scaling relationship between fluorescent intensity and polymer mass. Estimates of polymer mass obtained using electron tomography and live-cell imaging indicate that the tubulin within the spindle plateaus in early anaphase. This implies that the transverse density of microtubules within the anaphase B spindle decreases linearly with the length. Since, the transverse spindle width is below the resolving power of conventional epifluorescence microscopy, a useful proxy for tubulin density is the spindle's average fluorescent intensity, which indeed declines linearly with spindle length during anaphase B (*Figure 1A*).

The correspondence between fluorescent tubulin intensity and the mass of tubulin polymerised in the spindle can also be used to calculate the abundance and critical concentration of tubulin subunits within the fission yeast cell (*Table 1*). This calculation uses the well-defined structure of microtubules (*Howard, 2001*) to estimate the total number of tubulin subunits within the spindle. The intensity ratio between GFP-labelled tubulin in the spindle and cytoplasm can then be used to extrapolate this quantity to the abundance of tubulin subunits in the cytoplasm. Finally, the regular spherocylindrical shape of fission yeast cells, with a uniform radius and a well-defined length upon entry into mitosis, enables accurate estimates of the intracellular volume and thus concentration to be obtained. The estimates of tubulin abundance are in good agreement with mass-spectrometry studies (*Marguerat et al., 2012*), and yield an estimate for the critical tubulin concentration in fission yeast that is around 25% lower than the 4.9 ± 1.6 μM measured from in vitro tubulin polymerisation experiments (*Walker et al., 1988*; *Janson and Dogterom, 2004*).

## Sample preparation for electron tomography, acquisition and Processing

Yeast samples were prepared as in (*Roque et al., 2010*). Briefly, log yeast cultures were high pressure frozen using an EMPATC 2 (Leica Microsystems, Wetzlar, Germany) and fixed by freeze substitution with 0.1% dehydrated glutaraldehyde, 0.25% uranyl acetate and 0.01% osmium tretraoxide in dry-acetone. Freeze substitution was performed at −90°C for 56 hr after which the temperature was raised to −45°C in steps of 5°C/hr. Several washes of 15 m in dry-acetone were followed by lowicryl resin infiltration at −45°C in graded steps of 3:1, 1:1, 1:3 acetone:HM20 resin (EMS, cat. 14,340) for 1 hr each followed by 3 steps of 2 hr in 100% HM20 resin. After 100% HM20 over-night, fresh resin was added and polymerisation by UV light initiated while the samples were at −45°C and carried on while the samples were raised to 20°C in steps of 5°C/hr. The samples were then kept for 24 hr more at 20°C under UV light. Tilt-series of relevant areas were acquired from ±60° at 1° intervals in a Tecnai F30/F20/T12 (FEI, Oregon USA) at the magnification of 15,500×/14,500×/11,000×, respectively using the SerialEM software. Tilt-series were reconstructed, joined in the cases where spindles spanned several adjacent subsections, and tracked manually using the software package IMOD (*Kremer et al., 1996*).

In the electron tomograms, microtubules appear as cylindrical tubes with a diameter of around 18 nm compared with the 25 nm expected for 13-protofilament microtubules. The microtubule shrinkage is caused by the freeze-substitution methods used to fix the cells, but other aspects of the microtubule structure appeared unperturbed. All measurements were thus scaled uniformly by γ = 25/18 to obtain more accurate estimates of the spindles' physical properties.

## Geometric analysis of electron tomograms

The granularity of the isotropic fibre tracking analysis (IFTA) solution is controlled by a single parameter, $R_s$, which specifies the separation of neighbouring coordinates that define the axis of the spindle in three-dimensions. This parameter is set to 200 nm for tracking anaphase B spindles in fission yeast and at 100 nm for computing the histograms of nearest-neighbour distances and microtubule packing angles. The IFTA algorithm proceeds by first detecting the pole with largest number of microtubule minus-ends that lie within the threshold distance, $R_s$. The centre-of-mass of these ends is used as the first point defining the spindle. A sphere with a radius, $R_s$, is then centred at the pole, and used to detect the

positions where microtubules intersect the ball's surface. A constrained optimisation calculation is then used to determine the point lying on the sphere's surface that minimises the mean-squared distance from the microtubule crossings. This point is selected as the second position in the three-dimensional interpolation of the spindle axis whilst microtubule coordinates that lie within the sphere are masked. This procedure is repeated until the spindle coordinates reach the opposite spindle pole. The transverse organisation of the spindle is then determined by detecting the positions where microtubules intersect a plane halfway between and perpendicular to the points that define the spindle axis.

### Quantifying Microtubule organisation and the Transverse Stiffness of spindle

The calculation of the transverse stiffness is made by assuming that microtubules are uniform, hollow cylinders and that the cross-linkers are rectangular support elements with identical material properties. The parameters are taken from the known dimensions of microtubules and the microtubule numbers and organisation determined in this study. For a slender, elastic beam with length, L, constructed from a material with Young's modulus E, the compressive force that can be sustained before buckling is given by $F = \pi^2 EI/L^2$, provided that the two ends of the beam are allowed to pivot freely (*Landau et al., 1986*), as is likely to be the case for the fission yeast spindle (*Tolić-Nørrelykke, Sacconi, et al., 2004b*; *Kalinina et al., 2013*). Alternative boundary conditions lead the critical force to be altered by a multiplicative constant. The third property of the beam that determines the critical force is the area moment of inertia (*Figure 2*).

The contribution that a single microtubule with its centre a distance, y, from the neutral axis makes to the bundle's area moment of inertia in the y-direction, $I_{xx}$, is given by the following equation

$$\begin{aligned} I_{xx} &= \int_{area}(y + r\sin\theta)^2 dA = \int_{r_1}^{r_2}\left[y^2 + r^2\sin^2\theta + 2yr\sin\theta\right]r\,dr\,d\theta \\ &= \pi y^2\left(r_2^2 - r_1^2\right) + \frac{\pi}{4}\left(r_2^4 - r_1^4\right) \\ &= y^2 A_{MT} + I_{MT} \end{aligned}$$

(2)

where r is a radial coordinate representing distance from the microtubule axis, and $r_2$ and $r_1$ are the outer and inner radii of the cylindrical microtubule. The term $A_{MT}$ is the microtubule cross-sectional area and $I_{MT}$ is the (isotropic) moment of inertia of a single microtubule about its axis. A symmetric expression exists for the moment of inertia in the x-direction, and the product moment of inertia $I_{xy}$, is given by

$$I_{xy} = -\int_{area}(y + r\sin\theta)(x + r\cos\theta)dA = -\pi xy(r_2^2 - r_1^2) = -xyA_{MT}$$

(3)

The components of the moment of inertia tensor can be obtained by summing the contributions of individual microtubules. The position of the neutral axis about which the bundle of microtubules will undergo bending is given by the centre of mass of the microtubule centres, which gives the following expression for a bundle containing N microtubules with centres at $(x_i, y_i)$ with a mean positions $(\bar{x}, \bar{y})$ in the plane perpendicular to the bundle

$$\begin{aligned} I_{xx} &= NI_{MT} + A_{MT}\sum_{i=1}^{N}(y_i - \bar{y})^2 \\ I_{xy} &= -A_{MT}\sum_{i=1}^{N}(y_i - \bar{y})(x_i - \bar{x}) \end{aligned}$$

(4)

The formula for the area moment of inertia tensor, **J**, can thus be written compactly in matrix notation as

$$\mathbf{J} = \begin{pmatrix} I_{xx} & I_{xy} \\ I_{xy} & I_{yy} \end{pmatrix} = N\left[I_{MT}\mathbf{I}_2 + A_{MT}\left(tr\left(\Sigma\right)\mathbf{I}_2 - \Sigma\right)\right]$$

(5)

where $\mathbf{I}_2$ is the 2 × 2 identity matrix, tr(.) refers to the matrix trace operator and $\mathbf{\Sigma}$ is the covariance matrix of the microtubule centres in the xy-plane.

A similar expression can be obtained for a composite beam containing microtubules and cross-linkers, if it is assumed that the cross-linkers are rectangular support elements (with a width, w, and a height, h) and the same material properties as microtubules. However, since the rectangular

cross-linkers, themselves, have an anisotropic area moment of inertia, the first term in equation must also take into account the orientation, θ, of each cross-linker, which we define with respect to the y-axis as

$$J^{linker}(\theta) = \left(\frac{wh}{12}\right)\begin{pmatrix} h^2\cos^2\theta + w^2\sin^2\theta & (w^2 - h^2)\sin\theta\cos\theta \\ (w^2 - h^2)\sin\theta\cos\theta & h^2\sin^2\theta + w^2\cos^2\theta \end{pmatrix} \qquad (6)$$

## Effect of polymer mass conservation on the Spindle's Transverse Stiffness

The effect that polymer mass conservation has on the spindle's stiffness can be investigated by treating the spindle as a solid, homogeneous cylinder. If the volume of the cylinder, $V$, is held constant, to model the conservation of polymer mass, then the cylindrical beam becomes thinner as it elongates. The relationship between cylinder radius, $r_c$, and its length, $L_s$, ($r_c = \sqrt{V/\pi L_s}$) and the fourth-order scaling of transverse stiffness with beam radius, $EI \propto r_c^4$ (*Landau et al., 1986*) then lead to the $EI \propto L_s^{-2}$ scaling of transverse stiffness with length.

This scaling relation is consistent with the MMO model (*Figure 5*), which posits that the number of microtubules in a cross-section should be approximately $L_T/(L_s + L_m)$, because the transverse stiffness, $EI$, depends quadratically on the number of microtubules in a cross-section (*Figure 3C,D*).

## Computational models of yeast spindles

The response of the spindle to compressive forces was investigated using the cytoskeletal modelling software Cytosim, which solves the over-damped Langevin equations of cytoskeletal filaments using an implicit numerical integration scheme (*Nedelec and Foethke, 2007*). The code was compiled and run on the EMBL High Performance Computing cluster, with jobs submitted to the Platform LSF scheduler using custom Python scripts. Simulation results were analysed using Matlab (The MathWorks inc.).

The simulations were designed to reproduce the morphology and biophysical characteristics of each spindle sufficiently closely to estimate the spindle's critical force. The SPBs were modelled as cylindrical elastic solids, associated with a scalar drag coefficient. Each microtubule was connected to the SPB by a pair of Hookean springs. The first of these was coupled to the minus-end of the microtubule, and was given a large elastic constant to model the high resistance of wild-type SPBs to pushing forces (*Toya et al., 2007*). The steric exclusion between microtubules was implemented using a one-sided quadratic potential with a minimum at the steric radius of 30 nm (*Loughlin et al., 2010*).

## Estimating the critical force of ET-reconstructed spindle architectures

The number and length of microtubules in the models of spindles from wild-type fission yeast cells were determined directly from ET reconstructions, whilst the SPB separation (or spindle length) was set to the IFTA-derived contour length between the poles of the ET spindle. The lengths of microtubules in budding yeast and cdc25.22 fission yeast cells were measured from the line representations of each spindle in the respective publications (*Ding et al., 1993*; *Winey et al., 1995*). The budding yeast spindles contain short microtubules (with lengths less than 0.5 μm) that are likely to represent kinetochore fibres that were not fully depolymerised during anaphase A. These fibres are unlikely to contribute to the structural integrity of the spindle, and account for a small proportion (7.8 ± 4.7%, (S. D.)) of the total polymerised tubulin. These microtubules were therefore neglected in the spindle models. Two of the budding yeast spindles also contain pole-to-pole microtubules that cannot be unambiguously assigned to a specific spindle pole. In the first spindle (numbered 12 in Winey et al.) the two pole-to-pole microtubules are assigned symmetrically to each SPB. The second spindle (numbered 14 in Winey et al.) contains a single pole-to-pole microtubule that is assigned to the SPB with the lowest number of long nucleated microtubules.

Having determined the spindle length, the number of microtubules and their lengths, the positions of the microtubule minus-ends at each SPB are set by sampling a random position on the circular face of each SPB using a Monte Carlo method. Rejection sampling was first used to sample the area on the disk's surface with uniform probability. The Euclidean distance between a candidate point and the other microtubules was determined, and the point was only accepted if its separation was

greater than the twice the steric radius of each microtubule. This process was repeated until the SPB was populated with the correct number of microtubules. This procedure set the position of the microtubule minus-ends with respect to the SPB, with the position of plus-ends and the transverse organisation of microtubules at the midzone determined by simulating cross-linker attachment and detachment from the microtubule lattice. The SPBs were confined to the x-axis throughout the simulation to aid visualisation.

The simulation of microtubule organisation at the spindle midzone was performed using cross-linkers that only bind to pairs of anti-parallel microtubules (*Janson et al., 2007*). The cross-linkers were also confined to cylindrical region with a total length of 2.5 µm at the centre of the spindle, to approximate the width of the midzone in yeast cells (*Loïodice et al., 2005*; *Yamashita et al., 2005*). When bound to a pair of microtubules, cross-linkers behave as elastic bridging elements that set the centre-to-centre between pairs of microtubules at 50 nm. The stiffness of the cross-linkers is consistent with the known Young's modulus of the alpha-helical class of proteins to which the dimeric kinesins and Map65 proteins belong. The microtubules were also subjected to weak (20 Pa) centring forces to prevent rotational diffusion of the microtubules away from the spindle axis and thus ensure that cross-links were formed between the two halves of the bipolar spindle.

Throughout the initialisation of spindle architecture, the SPBs were connected to each other by a stiff spring with the same resting length as the spindle to prevent the SPB separation being altered by diffusion. After simulating the spindle for fifty seconds, almost all of the cross-linkers are bound to the midzone and the two halves of the spindle are strongly connected. Under these conditions, the cross-linkers are capable of forming the idealised square-packed arrays observed in yeast spindles, albeit with lower efficiency than we observe in electron tomogram reconstructions of wild-type spindles (data not shown). Upon completion of the initialisation step, the rate with which cross-linkers detach from the microtubule lattice was set to zero in order to probe the spindles' elastic response to increasing forces. This was effected by decreasing instantaneously the resting length of the spring connecting the two SPBs by 1 µm, and then linearly increasing the spring constant from a starting value of zero to a maximum of 240 pNµm$^{-1}$ over the remaining 150 s of the simulation. The stiffness of the elastic element was maintained at the maximal value for a further 50 s, during which time the critical force on the spindle was determined. This was carried out by averaging the forces borne by the elastic element connecting the pair of SPBs. In simulations of spindles with elastic reinforcement, each microtubule model point was confined by an elastic potential with a given degree of stiffness (*Figure 6A–D*). The stiffness of the spring compressing the spindle poles was also increased to a maximum value of 1500 pNµm$^{-1}$ to ensure that the critical force was exceeded, but all other simulation parameters were identical. Simulation parameters are provided in *Table 2*.

## Sampling Microtubule number and length using null statistical models

The simulations of the null models of spindle architecture were identical to those used to determine the critical force of wild-type spindles, except that microtubule length and number were sampled from probability distributions. The objective of this procedure was to sample a large number of alternative spindle morphologies to investigate the degree to which wild-type spindles are mechanically optimal. In constructing the null statistical models, the number of microtubules emanating from each SPB was sampled from a Poisson distribution with a mean equal to that observed in the wild-type spindle with the same length. The lengths of the microtubules in each random model were determined by randomly partitioning the total polymer present in the wild-type spindle between the $N_{MT}$ microtubules. In cases where the length of a sampled microtubule exceeded the spindle length, the microtubule was truncated to the length of the spindle with the remaining polymer used to set the length of one or more additional microtubules that were assigned to one of the two SPBs at random. This procedure increased slightly the average number of microtubules in the random spindles but ensured that the overall polymer mass was conserved. At SPBs that contained in excess of six microtubules, the radius of the circular face of the SPB was increased so that its area increased linearly with microtubule number, and that the density of microtubules on the surface of the SPB was constant.

## Estimating the forces on the anaphase B spindle

The viscous drag on the fission yeast nucleus is substantially larger than is predicted by Stokes' law due to the narrow separation between the nuclear envelope and the enclosing cell wall

(*Foethke et al., 2009*). The equation for translational motility of the nucleus in the cell geometry can be approximated as

$$\gamma = \frac{9\pi^2\sqrt{2}\eta_{cell}r_{nucleus}}{4\varepsilon^{5/2}}$$

where $\varepsilon = (r_{cell} - r_{nucleus})/r_{nucleus}$ is the cell clearance. A description of the other variables is shown in *Table 3*.

After ablation of the spindle midzone, the forces acting on the daughter nuclei are

$$F_{viscous} = -F_{elastic}$$

$$-\gamma\frac{dx}{dt} = kx$$

which has a solution, $x(t) = x_0 \cdot e^{(-t/\tau)}$, where $\tau = \gamma/k$ represents the characteristic time for the spindle to reach equilibrium, and $x_0$ is the initial displacement. An exponential fit to the relaxation data provides values for these variables, which can then also be combined to give a rough estimate of the total force resisting spindle elongation (*Table 3*).

## Acknowledgements

We thank Phong Tran for help in initiating the project. We thank Ken Sawin and Fred Chang for providing yeast strains, and Damian Brunner and members of his lab for yeast strains and technical help. We thank Carl Zeiss and PerkinElmer for continuous support of the EMBL Advanced Light Microscopy Facility. We would like to thank our colleagues Jan Ellenberg, Darren Gilmour, Marko Kaksonen, Robert Mahen, Marcel Janson and Jordan Raff for critical reading of the manuscript. Research in the Nedelec lab is supported by BioMS and the European Commission Seventh Framework Programme (FP7/2007-2013) under grant agreements no. 241548. and no. 258068. JJW acknowledges financial support from the VW foundation within the program; Computational Soft Matter and Biophysics (Grant No. I/83 942).

## Additional information

### Funding

| Funder | Grant reference number | Author |
|---|---|---|
| European Commission | 241548 and 258068 | Jonathan J Ward |
| Volkswagen Foundation | I/83 942 | Jonathan J Ward |

The funders had no role in study design, data collection and interpretation, or the decision to submit the work for publication.

### Author contributions

JJW, HR, Conception and design, Acquisition of data, Analysis and interpretation of data, Drafting or revising the article; CA, FN, Conception and design, Drafting or revising the article

## Additional files

### Supplementary file

• Source Code 1. Compressed plain text files containing raw data, scripts and analysis code. The directory 'EM_analysis' includes microtubule coordinates for eleven fission yeast spindles that were reconstructed using ET, in addition to MATLAB (The MathWorks inc.) programs for performing geometric analysis on the spindles' architecture. Scripts used to perform analyses of live-cell fluorescent imaging of fission yeast mitosis are included in the directory, 'LM_analysis'. The directory, 'simulation_configs' contains *Cytosim* configuration files for simulations of yeast spindles buckling under compressive force.

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
