## [Decision Letter]

Thank you for sending your work entitled “Mechanical optimality and scaling in the design of the mitotic spindle” for consideration at *eLife*. Your article has been favorably evaluated by Richard Losick (Senior editor) and 3 reviewers, one of whom is a member of our Board of Reviewing Editors.

The Reviewing editor and the other reviewers discussed their comments before we reached this decision, and the Reviewing editor has assembled the following comments to help you prepare a revised submission.

This interesting paper combines computer simulations and electron tomography to understand spindle mechanics. This is a very important topic, and the approach that the authors use is novel and exciting.

While EM structures of yeast spindles have been discussed before (Ding et al., J Cell Biol, 1993), the present work uses EM tomography, which is an important improvement. Also, the combination of mechanical models with EM data is novel and interesting. However, in the manuscript, it is not carefully explained what was known before and what the new contribution of the present work is. This needs to be better clarified.

Another more serious problem with the present manuscript is that it is, in many points, unclear and very difficult to read, in particular when it comes to the analysis of compressive strength. The main text is insufficient to understand key arguments. Repeatedly the terms “ideal” and “optimal” and “stereotyped” are used but are quite unclear. When following the referrals to figure captions and methods, it is often difficult to find some of the information needed. The figure captions in general are not fully satisfactory. All of these weaknesses of the presentation make the paper hard to read.

Below is a list of points that the authors should address in order to improve the manuscript.

1) The authors repeatedly state that the spindle is well ordered and has a stereotyped structure. I do not understand how Figure 1 supports those claims. Figure 1 shows that spindles with similar lengths have very different cross-sections, different degrees of microtubule overlap, and even different microtubule lengths. Thus, the spindles do not seem “stereotyped” to me. The cross-sections also look quite disordered, and Figure 1 shows very broad distribution of packing angles, thus, it seems to me that the spindles are not well ordered. It would be helpful if the authors tried to quantify the degree of order and the extent to which the spindles are stereotyped, perhaps by measuring a 2D crystalline order parameter.

2) In the Results section, the authors describe simulations comparing the buckling resistance of spindles with geometries measured from tomography to the buckling resistance of randomly generated arrays. This is a very interesting methodology. However, the conclusion that “This analysis also indicates that alternative spindle architectures with greater buckling resistance than is observed in wild-type cells do not appear to exist, and that the natural spindle architecture may be close to optimal” is still problematic, because it could depend on the null model that the authors chose. An alternative null model would be to generate random bundles of microtubules that would be expected to form with unregulated, non-specific crosslinkers. It would be very interesting for the authors to perform such simulations to test if there is really something special about the arrangement of microtubules in the spindle, or if the same resistance to buckling would result from unregulated crosslinked arrays of microtubules.

3) The tomography data in 1B shows that these three representative spindles have bends (this is most dramatic for the “early” spindle, but the bends are present in all the spindles). Since the spindles are already bent to some extent, then forces on the poles cannot cause a buckling (which would only occur from a starting straight configuration), but additional forces could cause further bending. Given this, the relevance of studying how a simulated straight spindle would buckle is unclear. It would be helpful for the authors to explain their reasoning in studying buckling. It would also be helpful for the authors to comment on the observed bending of the spindle in the tomograms and explain how it relates to their studies of the mechanics of bending of spindles.

4) The authors suggest that compressive strength is maximized. However, it does not become clear whether an optimal strength is needed to withstand the compressive forces during anaphase B. What is the compressive strength needed for the task? How many individual microtubules would be needed to withstand the compressive forces that arise? There is a discussion of this subject in the Discussion section, but it is unclear to me. 10 pN at the poles seem very small, and I do not understand why this “suggests that the precise regulation of spindle architecture (Figure 3) is essential for the spindle to exceed the required buckling tolerance in late anaphase B.” A clearer and more thorough discussion would be good. Also, the text following this sentence is unclear.

5) I do not understand the sentence: “This property is associated with the geometry of square and hexagonal arrays and applies to intact bundles irrespective of the cross-linker density” (in the Results section). I also do not see how Figure 2–figure supplement 1 should help to provide clarification. Where is cross-linker density defined and discussed?

6) The authors use the terms “transverse stiffness”, “buckling resistance”, “compressive strength”, “compressive force… before buckling”, and it is not clear to me if they all mean the same or if they refer to slightly different properties. It would be useful to have clear definitions of these terms and to clarify their relationship. Also, the use of “transverse stiffness” is confusing to me, as there are different transverse stiffnesses, and the authors seem only to refer to the minimal value. Switching from one term to another in the text is confusing as the reader is not sure whether a different spindle property is meant.

In this context, the following sentence, in the Results section, is completely unclear to me: “We also observed that the square-packed architecture, larger number of microtubules and increased microtubule separation at the spindle midzone result in a greater transverse stiffness than in the polar regions for all phases of spindle elongation.” To which figure does this statement refer?

7) Most mechanical arguments of the paper seem to rely on homogeneous elastic material properties. However, I did not find a discussion of this simplification and whether, in the case of microtubule bundles, this is a good simplification.

8) The sentence in the Results section, “Finally, we determined the buckling resistance of spindle architectures that maximise microtubule overlap at the stiff, square-packed spindle midzone, and are thus expected to possess optimal compressive strength”, is unclear.

How is overlap maximized? Does it mean overlap along the whole length? What does optimal mean? What is optimized under what conditions? Why is optimal put in “quotes”?

9) The repeated use of the terms “optimal” or “ideal” is unclear. It sounds like an exaggeration and the reader is left in the dark as to what exactly is meant.

10) In Materials and methods (“Quantifying microtubule organisation and the transverse stiffness of spindle”), when the buckling force F is introduced, the equation given corresponds to a beam with constant cross-section along the beam. This should be clarified. As the spindles in the present work do not have constant cross section, it would be useful to discuss in which way the simple law F=pi EI/L^2 is affected if cross-sections are not constant. This point could be used to motivate the numerical work discussed in the Results.

11) This paragraph, in the Results section, is unclear: “A comparison of the observed number of interpolar microtubules…” What does “precise scaling” mean here? What quantity is “scaling”? I also do not understand the following sentence in the same paragraph: “The theory also indicates that…”

---

## [Author Response]

As the reviewers mention in several of their comments, it is correct that any demonstration of spindle optimality is contingent on the model that is used, which, for a system as complex as the spindle, is always necessarily a simplification. We have therefore altered the tone of the article, so that we instead focus on the mechanical properties of the spindle that are likely to enhance its ability to generate force rather than on its optimality per se. This relatively small change in emphasis has involved changing the paper’s title and abstract but also resolves many of the major and minor criticisms identified by the reviewers. We have also revised the text extensively to improve both its clarity and precision, using a more precise terminology throughout.

Our revised manuscript includes two additional figures that respond to specific comments made by the reviewers. The first of these presents a more detailed description of the calculations of spindle stiffness. This is intended to provide the reader with a more intuitive picture of how the architecture of the spindle influences its mechanics. Although this information is provided as a figure, it could perhaps be better incorporated into the text as an explanatory “box” of the type found in review articles. We would be happy to receive guidance from the editorial team on this matter, considering that the necessary changes can easily be implemented by us. The second figure was added in response to one of the reviewers’ comments concerning the magnitude of the force that can be sustained by the spindle. This figure explores the effect of external elastic reinforcement on the forces that the wild-type spindle can support during its elongation.

As mentioned in the letter above, we have made extensive changes to the manuscript to improve its clarity and justify its major findings more carefully. We have also removed shorthand descriptions, such as “ideal”, “optimal” and other similar jargon wherever possible.

The contribution from Ding et al. is now referenced in the Introduction. The description of our wild-type electron tomograms now includes a comparison with the earlier serial-section reconstructions of cdc25.22 spindles (see “Similarities and differences between wild-type and cdc25.22 fission yeast spindle reconstructions”, in Results). The later theoretical sections of the paper explore the mechanical properties of both wild-type and cdc25.22 mutant spindle architectures, and provide novel insights into the organization of the spindle in both cell types. Comparisons of these two architectures provide insight into how the fission yeast spindle adapts to changes in cytoplasmic volume.

*1) The authors repeatedly state that the spindle is well ordered and has a stereotyped structure. I do not understand how*
Figure 1
*supports those claims.*
Figure 1
*shows that spindles with similar lengths have very different cross-sections, different degrees of microtubule overlap, and even different microtubule lengths. Thus, the spindles do not seem “stereotyped” to me*.

Indeed, the degree to which the spindles can be considered “stereotyped” is a somewhat subjective question that was not fully justified in the original manuscript. We have addressed this comment toning down the discussion of spindle “stereotypes” and instead draw attention to the regular features of the spindle that we do observe, such as the remarkable mirror symmetry between the poles in terms of microtubule number and length, and the structural motifs (2x2 square arrays at the midzone and triangles at the spindle poles) that are present in all of the anaphase B spindles. The later analyses of microtubule numbers and lengths, which should now be much more accessible to the reader, provide further evidence that these aspects of the spindle’s architecture are also under precise control.

*The cross-sections also look quite disordered, and*
Figure 1
*shows very broad distribution of packing angles, thus, it seems to me that the spindles are not well ordered. It would be helpful if the authors tried to quantify the degree of order and the extent to which the spindles are stereotyped, perhaps by measuring a 2D crystalline order parameter*.

The nearest-neighbor approach has been used extensively for measuring distances and packing angles (10; 83; 48), but does have several limitations. The most notable of these is associated with the angle measurements, which are defined with respect to the two nearest MTs. These triplets of MTs are not necessarily all from the same crystalline unit cell, which leads to a less uniform distribution of packing angles (and distances) being measured for the spindle. This effect partly explains the broad distribution of packing angles that are present, particularly in the earlier spindles. We agree that the transverse organization of cytoskeletal arrays is an area worthy of future study that would certainly benefit from the application of techniques from solid-state physics, but these techniques are relatively involved. We also feel that this quantitation would not provide a great deal of additional insight beyond the simple and intuitive nearest-neighbor method, and is therefore beyond the scope of this paper.

*2) In the Results section, the authors describe simulations comparing the buckling resistance of spindles with geometries measured from tomography to the buckling resistance of randomly generated arrays. This is a very interesting methodology. However, the conclusion that “This analysis also indicates that alternative spindle architectures with greater buckling resistance than is observed in wild-type cells do not appear to exist, and that the natural spindle architecture may be close to optimal” is still problematic, because it could depend on the null model that the authors chose. An alternative null model would be to generate random bundles of microtubules that would be expected to form with unregulated, non-specific crosslinkers. It would be very interesting for the authors to perform such simulations to test if there is really something special about the arrangement of microtubules in the spindle, or if the same resistance to buckling would result from unregulated crosslinked arrays of microtubules*.

This is an interesting suggestion, as the impact of cross-linker specificity on the organization of the spindle is currently unknown. It is also correct that the number of potential spindle organizations that could possibly exist is far larger than has been probed in the current study. However, these architectures are likely to be subject to other constraints on how they are formed and mediate elongation that cannot be probed using the simulation methodology that is described in the paper. The simulations were designed to approximate the organization of the spindle at a particular stage of its elongation without needing to simulate its progression through the earlier stages of mitosis. It is unclear whether the alternative architectures could ever be realized in vivo*.* We have responded to this comment by qualifying the discussion of this result more carefully. We have also amended the text to clarify the most important conclusion of this figure, which is to motivate and support the subsequent model of a maximal midzone-overlap (MMO) spindle architecture (this analysis also indicates that alternative spindle architectures with a larger critical force occur infrequently under this null model, and suggests that the lengths of wild-type microtubules are regulated to increase the spindle’s effective stiffness)*.*

*3) The tomography data in 1B shows that these three representative spindles have bends (this is most dramatic for the “early” spindle, but the bends are present in all the spindles). Since the spindles are already bent to some extent, then forces on the poles cannot cause a buckling (which would only occur from a starting straight configuration), but additional forces could cause further bending. Given this, the relevance of studying how a simulated straight spindle would buckle is unclear. It would be helpful for the authors to explain their reasoning in studying buckling. It would also be helpful for the authors to comment on the observed bending of the spindle in the tomograms and explain how it relates to their studies of the mechanics of bending of spindles*.

The bends in the spindle are an interesting feature of the electron tomographic reconstructions, and are indeed present in all of the anaphase B spindles. These deflections are more pronounced in the early anaphase B spindles, where they appear to be inconsistent with the linear spindle morphology that can be observed in living cells via light microscopy. As the reviewer also rightly suggests, it is unclear how a buckled spindle could elongate at such a uniform rate in the presence of variable resistive forces. These two lines of evidence suggest that the deflections are caused partially or entirely by the standard preparation of the cell for electron tomography and that the native spindles have a straighter morphology. Live-cell imaging studies suggest that the spindles in living cells are close to straight but that pathologically large compressive forces in mutant cells can lead to buckling and the eventual breakage of anaphase B spindles. These observations provide the rationale for studying the critical force of a spindle that is initially in a straight configuration.

It is likely that curved microtubules were also present in reconstructions of the cdc25.22 spindles, but due to the limitations of serial-section microscopy, these may have been artificially straightened or may have been impossible to reconstruct fully. Electron tomography remains the best technique that is available for studying cellular volumes of the size of the yeast spindle at high resolution. The advantage of ET over serial-section EM is that the full three-dimensional organization of the spindle can be recovered, whilst any distortions of the spindle can be corrected using the IFTA algorithm. This approach thus allows us to obtain spindle reconstructions across the full range of lengths that characterize anaphase B spindle elongation.

We have addressed this comment by discussing the deflections of the spindles observed with EM, including these arguments in “Similarities and differences between wild-type and cdc25.22 fission yeast spindle reconstructions”, in the Results section.

*4) The authors suggest that compressive strength is maximized. However, it does not become clear whether an optimal strength is needed to withstand the compressive forces during anaphase B. What is the compressive strength needed for the task? How many individual microtubules would be needed to withstand the compressive forces that arise? There is a discussion of this subject in the Discussion section, but it is unclear to me. 10 pN at the poles seem very small, and I do not understand why this “suggests that the precise regulation of spindle architecture (*Figure 3*) is essential for the spindle to exceed the required buckling tolerance in late anaphase B.” A clearer and more thorough discussion would be good. Also, the text following this sentence is unclear*.

We have introduced several paragraphs to the Discussion (“Spindle design principles and the generation of force”) that present estimates of the magnitude of the force on kinetochore microtubules in metaphase and in anaphase. These estimates suggest that the critical force of the metaphase spindle is greater than the force it is likely to bear during this phase of mitosis. Our calculations also suggest that the later anaphase B spindles can support the drag forces from the elongating nuclei. However, as the reviewer points out, this force seems intuitively relatively small, for example compared with the combined forces that the kinetochores could produce. We have thus introduced an additional figure, which shows that external mechanical reinforcement of the spindle could increase the force that the interpolar microtubules can support (in Results, “Elastic reinforcement may increase the forces that the spindle can sustain”). We have also added a corresponding paragraph to the Discussion *(*“However, the spindle may be subjected to larger forces…”) outlining the cellular mechanisms that could give rise to spindle reinforcement, and its significance for mitosis in fission yeast cells.

*5) I do not understand the sentence: “This property is associated with the geometry of square and hexagonal arrays and applies to intact bundles irrespective of the cross-linker density”, in the Results section. I also do not see how Figure 2–figure supplement 1 should help to provide clarification*. *Where is cross-linker density defined and discussed?*

In the original manuscript, this information was split between the Methods section and the main text, and admittedly was difficult to follow. We modeled increases in cross-linker density by increasing the width of the rectangular support elements between the microtubules. These calculations show that the plateaus in the “minimal transverse stiffness” (see below) are present whether the spindle is sparsely or densely cross-linked. We have amended the Results and Methods sections, in addition to the content and caption of the revised Figure 3 to explain this point more clearly.

*6) The authors use the terms “transverse stiffness”, “buckling resistance”, “compressive strength”, “compressive force… before buckling”, and it is not clear to me if they all mean the same or if they refer to slightly different properties. It would be useful to have clear definitions of these terms and to clarify their relationship. Also, the use of “transverse stiffness” is confusing to me, as there are different transverse stiffnesses, and the authors seem only to refer to the minimal value. Switching from one term to another in the text is confusing as the reader is not sure whether a different spindle property is meant*.

Apart from the Abstract, where we feel the more general and intuitive “compressive strength” is appropriate, we now use “critical force” consistently throughout the text, as this is the most accurate of the available terms. The new Figure 2 explains the concept of transverse stiffness as a 2D tensor in greater detail. For beams with a constant cross-section, the only quantity that contributes to the beams’ critical force is the small eigenvalue of the stiffness tensor, which we now refer to as “minimal transverse stiffness” to distinguish it from the tensor quantity.

*In this context, the following sentence, in the Results section, is completely unclear to me: “We also observed that the square-packed architecture, larger number of microtubules and increased microtubule separation at the spindle midzone result in a greater transverse stiffness than in the polar regions for all phases of spindle elongation*.*” To which figure does this statement refer?*

We have amended the text to refer to the relevant Figure 3. In the paragraph (“A notable feature of the stiffness of the idealised arrays…”), we now explain that the three properties that lead to the midzone having a greater stiffness than the pole regions include increased microtubule number, the lower intrinsic packing density of square arrays and the increased bridging distance between neighboring microtubules.

*7) Most mechanical arguments of the paper seem to rely on homogeneous elastic material properties. However, I did not find a discussion of this simplification and whether, in the case of microtubule bundles, this is a good simplification*.

We have added several paragraphs to the Discussion (“Material properties of the spindle”) that describe the conditions that lead to cross-linked cytoskeletal bundles behaving elastically, as is assumed in the calculations presented in Figure 3. The results of the numerical simulations, which are presented in subsequent figures, are not dependent on this assumption, as our algorithm considers each spindle as a composite structure that is not homogeneous. We, however, also provide arguments that the properties of the yeast spindle, in particular its slow dynamics, crystalline architecture and high abundance of crosslinkers, implies that the elastic assumption is a reasonable first approximation for this system.

8) The sentence in the Results section, “Finally, we determined the buckling resistance of spindle architectures that maximise microtubule overlap at the stiff, square-packed spindle midzone, and are thus expected to possess optimal compressive strength”, is unclear.

How is overlap maximized? Does it mean overlap along the whole length? What does optimal mean? What is optimized under what conditions? Why is optimal put in “quotes”?

The rationale for this model of spindle organization is now explained in greater detail in the manuscript (the overlap length is maximized, under two constraints: the total polymer mass L_T_ is given, and the pole-to-pole distance L_s_ is fixed). We have also added a diagram to Figure 5 (panels A and B) that explains how the overlap is maximised, given the constraints on polymer mass and the width of the midzone. As mentioned previously, we have significantly cut the jargon from the paper with a particular focus on any terms in “quotes”. In this case, we describe the spindle architecture in detail and refer to it as a maximal midzone-overlap (MMO) model, which is both precise and does not exaggerate its significance.

*9) The repeated use of the terms “optimal” or “ideal” is unclear. It sounds like an exaggeration and the reader is left in the dark as to what exactly is meant*.

We agree that the use of “optimal” and “ideal” as shorthand descriptions of different aspects of the spindle organization is a little vague, and we have tried to reduce the amount of jargon that is used in the paper as much as possible. In the revised manuscript, we now refer collectively to the triangular arrangement of microtubules at the poles of the spindle, and the 2x2 and 3x3 square arrays of microtubules at the midzone, as “structural motifs”. This term is both precise and does not imply anything for the structural motifs’ potential utility (see reply to major points 6 and 8).

*10) In Materials and methods (“Quantifying microtubule organisation and the transverse stiffness of spindle”), when the buckling force F is introduced, the equation given corresponds to a beam with constant cross-section along the beam. This should be clarified. As the spindles in the present work do not have constant cross section, it would be useful to discuss in which way the simple law F=pi EI/L^2 is affected if cross-sections are not constant. This point could be used to motivate the numerical work discussed in the Results*.

We have clarified the discussion of how the transverse stiffness influences a beam’s critical force in Figures 2 and 3. The expanded discussion of the stiffness tensor now explains how the anisotropy of a beam leads to deviations from simple, 1-dimensional buckling. We also note that the behavior of beams that do not have a constant cross-section is complex, which provides a valid justification for using simulations. We finally provide a more coherent link between the numerical simulations and the simple Euler-Bernoulli formula by computing the spindle’s effective stiffness E_eff_ I, from the critical forces measured in the simulation of the non-homogeneous spindles. This makes it clear to the reader that, although the two approaches differ greatly in their realism and complexity, they are in good agreement on how the spindle’s critical force scales with increasing pole separation.

11) This paragraph, in the Results section, is unclear: “A comparison of the observed number of interpolar microtubules…” What does “precise scaling” mean here? What quantity is “scaling”? I also do not understand the following sentence in the same paragraph: “The theory also indicates that…”

These two potential sources of confusion have largely been resolved by the changes made in response to major comment 8. We also have substantially expanded the caption of Figure 5 and the discussion of theoretical models of a maximally overlapping spindle to explain these results and their consequences more clearly.